# IT TAKES TWO TO TANGO: MIXUP FOR DEEP METRIC LEARNING

Shashanka Venkataramanan[1]*     Bill Psomas[3]*     Ewa Kijak[1]     Laurent Amsaleg[1]

Konstantinos Karantzalos[3]     Yannis Avrithis[2]

[1]Inria, Univ Rennes, CNRS, IRISA     [2]Athena RC

[3]National Technical University of Athens

## ABSTRACT

Metric learning involves learning a discriminative representation such that embeddings of similar classes are encouraged to be close, while embeddings of dissimilar classes are pushed far apart. State-of-the-art methods focus mostly on sophisticated loss functions or mining strategies. On the one hand, *metric learning losses* consider two or more examples at a time. On the other hand, modern *data augmentation* methods for *classification* consider two or more examples at a time. The combination of the two ideas is under-studied.

In this work, we aim to bridge this gap and improve representations using *mixup*, which is a powerful data augmentation approach interpolating two or more examples and corresponding target labels at a time. This task is challenging because unlike classification, the loss functions used in metric learning are not additive over examples, so the idea of interpolating target labels is not straightforward. To the best of our knowledge, we are the first to investigate mixing *both* examples and target labels for deep metric learning. We develop a generalized formulation that encompasses existing metric learning loss functions and modify it to accommodate for mixup, introducing *Metric Mix*, or *Metrix*. We also introduce a new metric—*utilization*—to demonstrate that by mixing examples during training, we are exploring areas of the embedding space beyond the training classes, thereby improving representations. To validate the effect of improved representations, we show that mixing inputs, intermediate representations or embeddings along with target labels significantly outperforms state-of-the-art metric learning methods on four benchmark deep metric learning datasets.

## 1 INTRODUCTION

*Classification* is one of the most studied tasks in machine learning and deep learning. It is a common source of pre-trained models for *transfer learning* to other tasks (Donahue et al., 2014; Kolesnikov et al., 2020). It has been studied under different *supervision settings* (Caron et al., 2018; Sohn et al., 2020), *knowledge transfer* (Hinton et al., 2015) and *data augmentation* (Cubuk et al., 2018), including the recent research on *mixup* (Zhang et al., 2018; Verma et al., 2019), where embeddings and labels are interpolated.

*Deep metric learning* is about learning from pairwise interactions such that inference relies on instance embeddings, *e.g.* for *nearest neighbor classification* (Oh Song et al.,

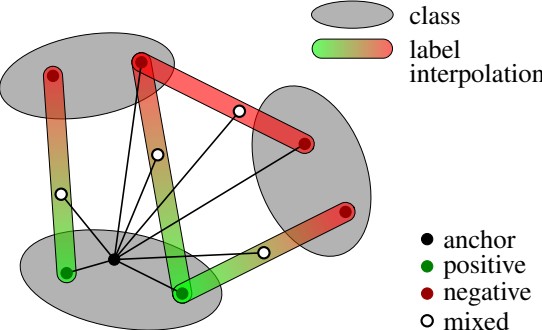

Figure 1: *Metrix* (= *Metric Mix*) allows an anchor to interact with positive (same class), negative (different class) and interpolated examples, which also have interpolated labels.

---

*equal contribution

2016), *instance-level retrieval* (Gordo et al., 2016), *few-shot learning* (Vinyals et al., 2016), *face recognition* (Schroff et al., 2015) and *semantic textual similarity* (Reimers & Gurevych, 2019).

Following (Xing et al., 2003), it is most often fully supervised by one class label per example, like classification. The two most studied problems are *loss functions* (Musgrave et al., 2020) and *hard example mining* (Wu et al., 2017; Robinson et al., 2021). Tuple-based losses with example weighting (Wang et al., 2019) can play the role of both.

Unlike classification, classes (and distributions) at training and inference are different in metric learning. Thus, one might expect interpolation-based data augmentation like mixup to be even more important in metric learning than in classification. Yet, recent attempts are mostly limited to special cases of embedding interpolation and have trouble with label interpolation (Ko & Gu, 2020). This raises the question: *what is a proper way to define and interpolate labels for metric learning?*

In this work, we observe that metric learning is not different from classification, where examples are replaced by pairs of examples and class labels by "positive" or "negative", according to whether class labels of individual examples are the same or not. The positive or negative label of an example, or a pair, is determined in relation to a given example which is called an *anchor*. Then, as shown in Figure 1, a straightforward way is to use a *binary* (two class) label per pair and interpolate it linearly as in standard mixup. We call our method *Metric Mix*, or *Metrix* for short.

To show that mixing examples improves representation learning, we quantitatively measure the properties of the test distributions using *alignment* and *uniformity* (Wang & Isola, 2020). *Alignment* measures the clustering quality and *uniformity* measures its distribution over the embedding space; a well clustered and uniformly spread distribution indicates higher representation quality. We also introduce a new metric, *utilization*, to measure the extent to which a test example, seen as a query, lies near any of the training examples, clean or mixed. By quantitatively measuring these three metrics, we show that interpolation-based data augmentation like mixup is very important in metric learning, given the difference between distributions at training and inference.

In summary, we make the following contributions:

1. We define a generic way of representing and interpolating labels, which allows straightforward extension of any kind of mixup to deep metric learning for a large class of loss functions. We develop our method on a generic formulation that encapsulates these functions (section 3).

2. We define the "positivity" of a mixed example and we study precisely how it increases as a function of the interpolation factor, both in theory and empirically (subsection 3.6).

3. We systematically evaluate mixup for deep metric learning under different settings, including mixup at different representation levels (input/manifold), mixup of different pairs of examples (anchors/positives/negatives), loss functions and hard example mining (subsection 4.2).

4. We introduce a new evaluation metric, *utilization*, validating that a representation more appropriate for test classes is implicitly learned during exploration of the embedding space in the presence of mixup (subsection 4.3).

5. We improve the state of the art on four common metric learning benchmarks (subsection 4.2).

## 2  RELATED WORK

**Metric learning**  Metric learning aims to learn a metric such that *positive* pairs of examples are nearby and *negative* ones are far away. In *deep metric learning*, we learn an explicit non-linear mapping from raw input to a low-dimensional *embedding space* (Oh Song et al., 2016), where the Euclidean distance has the desired properties. Although learning can be unsupervised (Hadsell et al., 2006), deep metric learning has mostly followed the supervised approach, where positive and negative pairs are defined as having the same or different class label, respectively (Xing et al., 2003).

Loss functions can be distinguished into pair-based and proxy-based (Musgrave et al., 2020). *Pair-based* losses use pairs of examples (Wu et al., 2017; Hadsell et al., 2006), which can be defined over triplets (Wang et al., 2014; Schroff et al., 2015; Weinberger & Saul, 2009; Hermans et al., 2017), quadruples (Chen et al., 2017) or tuples (Sohn, 2016; Oh Song et al., 2016; Wang et al., 2019). *Proxy-based* losses use one or more proxies per class, which are learnable parameters in the embedding space (Movshovitz-Attias et al., 2017; Qian et al., 2019; Kim et al., 2020c; Teh et al.,

2020; Zhu et al., 2020b). Pair-based losses capture data-to-data relations, but they are sensitive to noisy labels and outliers. They often involve terms where given constraints are satisfied, which produce zero gradients and do not contribute to training. This necessitates *mining* of hard examples that violate the constraints, like semi-hard (Schroff et al., 2015) and distance weighted (Wu et al., 2017). By contrast, proxy-based losses use data-to-proxy relations, assuming proxies can capture the global structure of the embedding space. They involve less computations that are more likely to produce nonzero gradient, hence have less or no dependence on mining and converge faster.

**Mixup**   *Input mixup* (Zhang et al., 2018) linearly interpolates between two or more examples in the input space for data augmentation. Numerous variants take advantage of the structure of the input space to interpolate non-linearly, *e.g.* for images (Yun et al., 2019; Kim et al., 2020a; 2021; Hendrycks et al., 2020; DeVries & Taylor, 2017; Qin et al., 2020; Uddin et al., 2021). *Manifold mixup* (Verma et al., 2019) interpolates intermediate representations instead, where the structure is learned. This can be applied to or assisted by decoding back to the input space (Berthelot et al., 2018; Liu et al., 2018; Beckham et al., 2019; Zhu et al., 2020a; Venkataramanan et al., 2021). In both cases, corresponding labels are linearly interpolated too. Most studies are limited to cross-entropy loss for classification. Pairwise loss functions have been under-studied, as discussed below.

**Interpolation for pairwise loss functions**   As discussed in subsection 3.3, interpolating target labels is not straightforward in pairwise loss functions. In *deep metric learning*, *embedding expansion* (Ko & Gu, 2020), HDML (Zheng et al., 2019) and *symmetrical synthesis* (Gu & Ko, 2020) interpolate pairs of embeddings in a deterministic way within the same class, applying to pair-based losses, while *proxy synthesis* (Gu et al., 2021) interpolates between classes, applying to proxy-based losses. None performs label interpolation, which means that (Gu et al., 2021) risks synthesizing false negatives when the interpolation factor $\lambda$ is close to 0 or 1.

In *contrastive representation learning*, MoCHi (Kalantidis et al., 2020) interpolates anchor with negative embeddings but not labels and chooses $\lambda \in [0, 0.5]$ to avoid false negatives. This resembles thresholding of $\lambda$ at 0.5 in OptTransMix (Zhu et al., 2020a). Finally, *i-mix* (Lee et al., 2021) and MixCo (Kim et al., 2020b) interpolate pairs of anchor embeddings as well as their (virtual) class labels linearly. There is only one positive, while all negatives are clean, so it cannot take advantage of interpolation for relative weighting of positives/negatives per anchor (Wang et al., 2019).

By contrast, Metrix is developed for deep metric learning and applies to a large class of both pair-based and proxy-based losses. It can interpolate inputs, intermediate features or embeddings of anchors, (multiple) positives or negatives *and* the corresponding two-class (positive/negative) labels per anchor, such that relative weighting of positives/negatives depends on interpolation.

## 3   MIXUP FOR METRIC LEARNING

### 3.1   PRELIMINARIES

**Problem formulation**   We are given a training set $X \subset \mathcal{X}$, where $\mathcal{X}$ is the input space. For each *anchor* $a \in X$, we are also given a set $P(a) \subset X$ of *positives* and a set $N(a) \subset X$ of *negatives*. The positives are typically examples that belong to the same class as the anchor, while negatives belong to a different class. The objective is to train the parameters $\theta$ of a model $f : X \rightarrow \mathbb{R}^d$ that maps input examples to a $d$-dimensional *embedding*, such that positives are close to the anchor and negatives are far away in the embedding space. Given two examples $x, x' \in \mathcal{X}$, we denote by $s(x, x')$ the *similarity* between $x, x'$ in the embedding space, typically a decreasing function of Euclidean distance. It is common to $\ell_2$-normalize embeddings and define $s(x, x') := \langle f(x), f(x') \rangle$, which is the *cosine similarity*. To simplify notation, we drop the dependence of $f, s$ on $\theta$.

*Pair-based* losses (Hadsell et al., 2006; Wang et al., 2014; Oh Song et al., 2016; Wang et al., 2019) use both anchors and positives/negatives in $X$, as discussed above. *Proxy-based* losses define one or more learnable *proxies* $\in \mathbb{R}^d$ per class, and only use proxies as anchors (Kim et al., 2020c) or as positives/negatives (Movshovitz-Attias et al., 2017; Qian et al., 2019; Teh et al., 2020). To accommodate for uniform exposition, we extend the definition of similarity as $s(v, x) := \langle v, f(x) \rangle$ for $v \in \mathbb{R}^d, x \in \mathcal{X}$ (proxy anchors) and $s(x, v) := \langle f(x), v \rangle$ for $x \in \mathcal{X}, v \in \mathbb{R}^d$ (proxy positives/negatives). Finally, to accommodate for mixed embeddings in subsection 3.5, we define $s(v, v') := \langle v, v' \rangle$ for $v, v' \in \mathbb{R}^d$. Thus, we define $s : (\mathcal{X} \cup \mathbb{R}^d)^2 \rightarrow \mathbb{R}$ over pairs of either

inputs in $\mathcal{X}$ or embeddings in $\mathbb{R}^d$. We discuss a few representative loss functions below, before deriving a generic form.

**Contrastive** The contrastive loss (Hadsell et al., 2006) encourages positive examples to be pulled towards the anchor and negative examples to be pushed away by a margin $m \in \mathbb{R}$. This loss is *additive* over positives and negatives, defined as:

$$\ell_{\text{cont}}(a; \theta) := \sum_{p \in P(a)} -s(a, p) + \sum_{n \in N(a)} [s(a, n) - m]_+. \tag{1}$$

**Multi-Similarity** The multi-similarity loss (Wang et al., 2019) introduces *relative weighting* to encourage positives (negatives) that are farthest from (closest to) the anchor to be pulled towards (pushed away from) the anchor by a higher weight. This loss is *not* additive over positives and negatives:

$$\ell_{\text{MS}}(a; \theta) := \frac{1}{\beta} \log \left( 1 + \sum_{p \in P(a)} e^{-\beta(s(a,p)-m)} \right) + \frac{1}{\gamma} \log \left( 1 + \sum_{n \in N(a)} e^{\gamma(s(a,n)-m)} \right). \tag{2}$$

Here, $\beta, \gamma \in \mathbb{R}$ are scaling factors for positives, negatives respectively.

**Proxy Anchor** The proxy anchor loss (Kim et al., 2020c) defines a learnable *proxy* in $\mathbb{R}^d$ for each class and only uses proxies as anchors. For a given anchor (proxy) $a \in \mathbb{R}^d$, the loss has the same form as (2), although similarity $s$ is evaluated on $\mathbb{R}^d \times \mathcal{X}$.

## 3.2 GENERIC LOSS FORMULATION

We observe that both additive (1) and non-additive (2) loss functions involve a sum over positives $P(a)$ and a sum over negatives $N(a)$. They also involve a decreasing function of similarity $s(a, p)$ for each positive $p \in P(a)$ and an increasing function of similarity $s(a, n)$ for each negative $n \in N(a)$. Let us denote by $\rho^+, \rho^-$ this function for positives, negatives respectively. Then, non-additive functions differ from additive by the use of a nonlinear function $\sigma^+, \sigma^-$ on positive and negative terms respectively, as well as possibly another nonlinear function $\tau$ on their sum:

$$\ell(a; \theta) := \tau \left( \sigma^+ \left( \sum_{p \in P(a)} \rho^+(s(a, p)) \right) + \sigma^- \left( \sum_{n \in N(a)} \rho^-(s(a, n)) \right) \right). \tag{3}$$

With the appropriate choice for $\tau, \sigma^+, \sigma^-, \rho^+, \rho^-$, this definition encompasses contrastive (1), multi-similarity (2) or proxy-anchor as well as many pair-based or proxy-based loss functions, as shown in Table 1. It does not encompass the *triplet loss* (Wang et al., 2014), which operates on pairs of positives and negatives, forming triplets with the anchor. The triplet loss is the most challenging in terms of mining because there is a very large number of pairs and only few contribute to the loss. We only use function $\tau$ to accommodate for *lifted structure* (Oh Song et al., 2016; Hermans et al., 2017), where $\tau(x) := [x]_+$ is reminiscent of the triplet loss. We observe that multi-similarity (Wang et al., 2019) differs from *binomial deviance* (Yi et al., 2014) only in the weights of the positive and negative terms. Proxy anchor (Kim et al., 2020c) is a proxy version of multi-similarity (Wang et al., 2019) on anchors and ProxyNCA (Movshovitz-Attias et al., 2017) is a proxy version of NCA (Goldberger et al., 2005) on positives/negatives.

This generic formulation highlights the components of the loss functions that are additive over positives/negatives and paves the way towards incorporating mixup.

## 3.3 IMPROVING REPRESENTATIONS USING MIXUP

To improve the learned representations, we follow (Zhang et al., 2018; Verma et al., 2019) in mixing inputs and features from intermediate network layers, respectively. Both are developed for classification.

*Input mixup* (Zhang et al., 2018) augments data by linear interpolation between a pair of input examples. Given two examples $x, x' \in \mathcal{X}$ we draw $\lambda \sim \text{Beta}(\alpha, \alpha)$ as *interpolation factor* and mix $x$ with $x'$ using the standard mixup operation $\text{mix}_\lambda(x, x') := \lambda x + (1 - \lambda)x'$.

| LOSS | ANCHOR | POS/NEG | $\tau(x)$ | $\sigma^+(x)$ | $\sigma^-(x)$ | $\rho^+(x)$ | $\rho^-(x)$ |
|---|---|---|---|---|---|---|---|
| Contrastive (Hadsell et al., 2006) | $X$ | $X$ | $x$ | $x$ | $x$ | $-x$ | $[x-m]_+$ |
| Lifted structure (Hermans et al., 2017) | $X$ | $X$ | $[x]_+$ | $\log(x)$ | $\log(x)$ | $e^{-x}$ | $e^{x-m}$ |
| Binomial deviance (Yi et al., 2014) | $X$ | $X$ | $x$ | $\log(1+x)$ | $\log(1+x)$ | $e^{-\beta(x-m)}$ | $e^{\gamma(x-m)}$ |
| Multi-similarity (Wang et al., 2019) | $X$ | $X$ | $x$ | $\frac{1}{\beta}\log(1+x)$ | $\frac{1}{\gamma}\log(1+x)$ | $e^{-\beta(x-m)}$ | $e^{\gamma(x-m)}$ |
| Proxy anchor (Kim et al., 2020c) | proxy | $X$ | $x$ | $\frac{1}{\beta}\log(1+x)$ | $\frac{1}{\gamma}\log(1+x)$ | $e^{-\beta(x-m)}$ | $e^{\gamma(x-m)}$ |
| NCA (Goldberger et al., 2005) | $X$ | $X$ | $x$ | $-\log(x)$ | $\log(x)$ | $e^x$ | $e^x$ |
| ProxyNCA (Movshovitz-Attias et al., 2017) | $X$ | proxy | $x$ | $-\log(x)$ | $\log(x)$ | $e^x$ | $e^x$ |
| ProxyNCA++ (Teh et al., 2020) | $X$ | proxy | $x$ | $-\log(x)$ | $\log(x)$ | $e^{x/T}$ | $e^{x/T}$ |

Table 1: Loss functions. Anchor/positive/negative: $X$: embedding of input example from training set $X$ by $f$; proxy: learnable parameter in $\mathbb{R}^d$; $T$: temperature. All loss functions are encompassed by (3) using the appropriate definition of functions $\tau, \sigma^+, \sigma^-, \rho^+, \rho^-$ as given here.

*Manifold mixup* (Verma et al., 2019) linearly interpolates between intermediate representations (features) of the network instead. Referring to 2D images, we define $g_m : \mathcal{X} \to \mathbb{R}^{c \times w \times h}$ as the mapping from the input to intermediate layer $m$ of the network and $f_m : \mathbb{R}^{c \times w \times h} \to \mathbb{R}^d$ as the mapping from intermediate layer $m$ to the embedding, where $c$ is the number of channels (feature dimensions) and $w \times h$ is the spatial resolution. Thus, our model $f$ can be expressed as the composition $f = f_m \circ g_m$.

For manifold mixup, we follow (Venkataramanan et al., 2021) and mix either features of intermediate layer $m$ or the final embeddings. Thus, we define three *mixup types* in total:

$$f_\lambda(x, x') := \begin{cases} f(\text{mix}_\lambda(x, x')), & \text{input mixup} \\ f_m(\text{mix}_\lambda(g_m(x), g_m(x'))), & \text{feature mixup} \\ \text{mix}_\lambda(f(x), f(x')), & \text{embedding mixup.} \end{cases} \quad (4)$$

Function $f_\lambda : \mathcal{X}^2 \to \mathbb{R}^d$ performs both mixup and embedding. We explore different mixup types in subsection B.4 of the Appendix.

### 3.4 LABEL REPRESENTATION

**Classification** In supervised classification, each example $x \in X$ is assigned an one-hot encoded label $y \in \{0,1\}^C$, where $C$ is the number of classes. Label vectors are also linearly interpolated: given two labeled examples $(x, y), (x', y')$, the interpolated label is $\text{mix}_\lambda(y, y')$. The loss (cross-entropy) is a continuous function of the label vector. We extend this idea to metric learning.

**Metric learning** Positives $P(a)$ and negatives $N(a)$ of anchor $a$ are defined as having the same or different class label as the anchor, respectively. To every example in $P(a) \cup N(a)$, we assign a binary (two-class) label $y \in \{0,1\}$, such that $y = 1$ for positives and $y = 0$ for negatives:

$$U^+(a) := \{(p, 1) : p \in P(a)\} \quad (5)$$

$$U^-(a) := \{(n, 0) : n \in N(a)\} \quad (6)$$

Thus, we represent both positives and negatives by $U(a) := U^+(a) \cup U^-(a)$. We now rewrite the generic loss function (3) as:

$$\ell(a; \theta) := \tau \left( \sigma^+ \left( \sum_{(x,y) \in U(a)} y \rho^+(s(a, x)) \right) + \sigma^- \left( \sum_{(x,y) \in U(a)} (1 - y) \rho^-(s(a, x)) \right) \right). \quad (7)$$

Here, every labeled example $(x, y)$ in $U(a)$ appears in both positive and negative terms. However, because label $y$ is binary, only one of the two contributions is nonzero. Now, in the presence of mixup, we can linearly interpolate labels exactly as in classification.

### 3.5 MIXED LOSS FUNCTION

**Mixup** For every anchor $a$, we are given a set $M(a)$ of pairs of examples to mix. This is a subset of $(S(a) \cup U(a)) \times U(a)$ where $S(a) := (a, 1)$. That is, we allow mixing between positive-negative, positive-positive and negative-negative pairs, where the anchor itself is also seen as positive. We

define the possible choices of *mixing pairs* $M(a)$ in subsection 4.1 and we assess them in subsection B.4 of the Appendix. Let $V(a)$ be the set of corresponding *labeled mixed embeddings*

$$V(a) := \{(f_\lambda(x, x'), \text{mix}_\lambda(y, y')) : ((x, y), (x', y')) \in M(a), \lambda \sim \text{Beta}(\alpha, \alpha)\}, \quad (8)$$

where $f_\lambda$ is defined by (4). With these definitions in place, the generic loss function $\widetilde{\ell}$ over mixed examples takes exactly the same form as (7), with only $U(a)$ replaced by $V(a)$:

$$\widetilde{\ell}(a; \theta) := \tau \left( \sigma^+ \left( \sum_{(v, y) \in V(a)} y \rho^+(s(a, v)) \right) + \sigma^- \left( \sum_{(v, y) \in V(a)} (1 - y) \rho^-(s(a, v)) \right) \right), \quad (9)$$

where similarity $s$ is evaluated on $\mathcal{X} \times \mathbb{R}^d$ for pair-based losses and on $\mathbb{R}^d \times \mathbb{R}^d$ for proxy anchor. Now, every labeled embedding $(v, y)$ in $V(a)$ appears in both positive and negative terms and *both* contributions are nonzero for positive-negative pairs, because after interpolation, $y \in [0, 1]$.

**Error function**    Parameters $\theta$ are learned by minimizing the error function, which is a linear combination of the *clean loss* (3) and the *mixed loss* (9), averaged over all anchors

$$E(X; \theta) := \frac{1}{|X|} \sum_{a \in X} \ell(a; \theta) + w\widetilde{\ell}(a; \theta), \quad (10)$$

where $w \geq 0$ is the *mixing strength*. At least for manifold mixup, this combination comes at little additional cost, since clean embeddings are readily available.

### 3.6    Analysis: Mixed embeddings and positivity

Let $\text{Pos}(a, v)$ be the event that a mixed embedding $v$ behaves as "positive" for anchor $a$, *i.e.*, minimizing the loss $\widetilde{\ell}(a; \theta)$ will increase the similarity $s(a, v)$. In subsection A.2 of the Appendix, we explain that this "positivity" is equivalent to $\partial \widetilde{\ell}(a; \theta) / \partial s(a, v) \leq 0$. Under positive-negative mixing, *i.e.*, $M(a) \subset U^+(a) \times U^-(a)$, we then estimate the probability of $\text{Pos}(a, v)$ as a function of $\lambda$ in the case of multi-similarity (2) with a single mixed embedding $v$:

$$\text{P}(\text{Pos}(a, v)) = F_\lambda \left( \frac{1}{\beta + \gamma} \ln \left( \frac{\lambda}{1 - \lambda} \right) + m \right), \quad (11)$$

where $F_\lambda$ is the CDF of similarities $s(a, v)$ between anchors $a$ and mixed embeddings $v$ with interpolation factor $\lambda$. In Figure 2, we measure the probability of $\text{Pos}(a, v)$ as a function of $\lambda$ in two ways, both purely empirically and theoretically by (11). Both measurements are increasing functions of $\lambda$ of sigmoidal shape, where a mixed embedding is mostly positive for $\lambda$ close to 1 and mostly negative for $\lambda$ close to 0.

## 4    Experiments

### 4.1    Setup

**Datasets**    We experiment on Caltech-UCSD Birds (CUB200) (Wah et al., 2011), Stanford Cars (Cars196) (Krause et al., 2013), Stanford Online Products (SOP) (Oh Song et al., 2016) and In-Shop Clothing retrieval (In-Shop) (Liu et al., 2016) image datasets. More details are in subsection B.1.

**Network, features and embeddings**    We use Resnet-50 (He et al., 2016) (R-50) pretrained on ImageNet (Russakovsky et al., 2015) as a backbone network. We obtain the intermediate representation (*feature*), a $7 \times 7 \times 2048$ tensor, from the last convolutional layer. Following (Kim et al., 2020c), we combine adaptive average pooling with max pooling, followed by a fully-connected layer to obtain the *embedding* of $d = 512$ dimensions.

**Loss functions**    We reproduce *contrastive* (Cont) (Hadsell et al., 2006), *multi-similarity* (MS) (Wang et al., 2019), *proxy anchor* (PA) (Kim et al., 2020c) and *ProxyNCA++* (Teh et al., 2020) and we evaluate them under different mixup types. For MS (2), following Musgrave et al. (2020), we use $\beta = 18$, $\gamma = 75$ and $m = 0.77$. For PA, we use $\beta = \gamma = 32$ and $m = 0.1$, as reported by the authors. Details on training are in subsection B.1 of the Appendix.

**Methods**   We compare our method, *Metrix*, with *proxy synthesis* (PS) (Gu et al., 2021), *i-mix* (Lee et al., 2021) and MoCHi (Kalantidis et al., 2020). For PS, we adapt the official code[1] to PA on all datasets, and use it with PA only, because it is designed for proxy-based losses. PS has been shown superior to (Ko & Gu, 2020; Gu & Ko, 2020), although in different networks. MoCHi and *i-mix* are meant for contrastive representation learning. We evaluate using Recall@$K$ (Oh Song et al., 2016): For each test example taken as a query, we find its $K$-nearest neighbors in the test set excluding itself in the embedding space. We assign a score of $1$ if an example of the same class is contained in the neighbors and $0$ otherwise. Recall@$K$ is the average of this score over the test set.

**Mixup settings**   For input mixup, we use the $k$ *hardest negative* examples for each anchor (each example in the batch) and mix them with positives or with the anchor. We use $k = 3$ by default. For manifold mixup, we focus on the *last* few layers instead, where features and embeddings are compact, and we mix all pairs. We use feature mixup by default and call it *Metrix/feature* or just *Metrix*, while input and embedding mixup are called *Metrix/input* and *Metrix/embed*, respectively. For all mixup types, we use clean examples as anchors and we define a set $M(a)$ of pairs of examples to mix for each anchor $a$, with their labels (positive or negative). By default, we mix positive-negative or anchor-negative pairs, by choosing uniformly at random between $M(a) := U^+(a) \times U^-(a)$ and $M(a) := S(a) \times U^-(a)$, respectively, where $U^-(a)$ is replaced by hard negatives only for input mixup. More details are in subsection B.2.

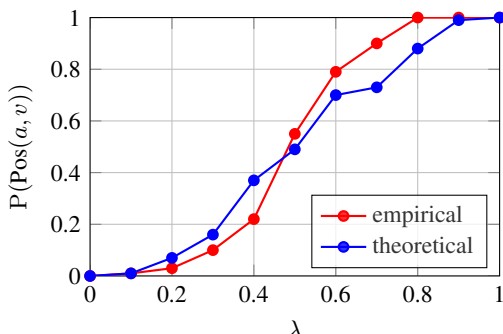

Figure 2: *"Positivity" of mixed embeddings* vs. $\lambda$. We measure $\mathrm{P}(\mathrm{Pos}(a,v))$ empirically as $\mathrm{P}(\partial \widetilde{\ell}_{\mathrm{MS}}(a;\theta)/\partial s(a,v) \leq 0)$ and theoretically by (11), where $F_\lambda$ is again measured from data. We use embedding mixup on MS (2) on CUB200 at epoch 0, based on the setup of subsection 4.1.

## 4.2   RESULTS

**Improving the state of the art**   As shown in Table 2, Metrix consistently improves the performance of all baseline losses (Cont, MS, PA, ProxyNCA++) across all datasets. More results in subsection B.3 of the Appendix reveal that the same is true for Metrix/input and Metrix/embed too. Surprisingly, MS outperforms PA and ProxyNCA++ under mixup on all datasets but SOP, where the three losses are on par. This is despite the fact that baseline PA outperforms MS on CUB200 and Cars-196, while ProxyNCA++ outperforms MS on SOP and In-Shop. Both contrastive and MS are significantly improved by mixup. By contrast, improvements on PA and ProxyNCA++ are marginal, which may be due to the already strong performance of PA, or further improvement is possible by employing different mixup methods that take advantage of the image structure.

In terms of Recall@1, our MS+Metrix is best overall, improving by 3.6% (67.8 → 71.4) on CUB200, 1.8% (87.8 → 89.6) on Cars196, 4.1% (76.9 → 81.0) on SOP and 2.1% (90.1 → 92.2) on In-Shop. The same solution sets new state of the art, outperforming the previously best PA by 1.7% (69.7 → 71.4) on CUB200, MS by 1.8% (87.8 → 89.6) on Cars196, ProxyNCA++ by 0.3% (80.7 → 81.0) on SOP and SoftTriple by 1.2% (91.0 → 92.2) on In-Shop. Importantly, while the previous state of the art comes from a different loss per dataset, MS+Metrix is almost consistently best across all datasets.

**Alternative mixing methods**   In Table 3, we compare Metrix/input with $i$-Mix (Lee et al., 2021) and Metrix/embed with MoCHi (Kalantidis et al., 2020) using contrastive loss, and with PS (Gu et al., 2021) using PA. MoCHi and PS mix embeddings only, while labels are always negative. For $i$-Mix, we mix anchor-negative pairs ($S(a) \times U^-(a)$). For MoCHi, the anchor is clean and we mix negative-negative ($U^-(a)^2$) and anchor-negative ($S(a) \times U^-(a)$) pairs, where $U^-(a)$ is replaced by $k = 100$ hardest negatives and $\lambda \in (0, 0.5)$ for anchor-negative. PS mixes embeddings of different classes and treats them as new classes. For clean anchors, this corresponds to positive-negative ($U^+(a) \times U^-(a)$) and negative-negative ($U^-(a)^2$) pairs, but PS also supports mixed anchors.

---

[1] https://github.com/navervision/proxy-synthesis

| METHOD | CUB200 | | | CARS196 | | | SOP | | | IN-SHOP | | |
|---|---|---|---|---|---|---|---|---|---|---|---|---|
| | R@1 | R@2 | R@4 | R@1 | R@2 | R@4 | R@1 | R@10 | R@100 | R@1 | R@10 | R@20 |
| Triplet (Weinberger & Saul, 2009) | 63.5 | 75.6 | 84.4 | 77.3 | 85.4 | 90.8 | 70.5 | 85.6 | 94.3 | 85.3 | 96.6 | 97.8 |
| LiftedStructure (Oh Song et al., 2016) | 65.9 | 75.8 | 84.5 | 81.4 | 88.3 | 92.4 | 76.1 | 88.6 | 95.2 | 88.6 | 97.6 | 98.4 |
| ProxyNCA (Movshovitz-Attias et al., 2017) | 65.2 | 75.6 | 83.8 | 81.2 | 87.9 | 92.6 | 73.2 | 87.0 | 94.4 | 86.2 | 95.9 | 97.0 |
| Margin (Wu et al., 2017) | 65.0 | 76.2 | 84.6 | 82.1 | 88.7 | 92.7 | 74.8 | 87.8 | 94.8 | 88.6 | 97.0 | 97.8 |
| SoftTriple (Qian et al., 2019) | 67.3 | 77.7 | 86.2 | 86.5 | 91.9 | 95.3 | 79.8 | 91.2 | 96.3 | **91.0** | 97.6 | 98.3 |
| D&C (Sanakoyeu et al., 2019)* | 65.9 | 76.6 | 84.4 | 84.6 | 90.7 | 94.1 | 75.9 | 88.4 | 94.9 | 85.7 | 95.5 | 96.9 |
| EPSHN (Xuan et al., 2020)* | 64.9 | 75.3 | 83.5 | 82.7 | 89.3 | 93.0 | 78.3 | 90.7 | 96.3 | 87.8 | 95.7 | 96.8 |
| Cont (Hadsell et al., 2006) | 64.7 | 75.9 | 84.6 | 81.6 | 88.2 | 92.7 | 74.9 | 87.0 | 93.9 | 86.4 | 94.7 | 96.2 |
| +Metrix | 67.4 | 77.9 | 85.7 | 85.1 | 91.1 | 94.6 | 77.5 | 89.1 | 95.5 | 89.1 | 95.7 | 97.1 |
| | +2.7 | +2.0 | +1.1 | +3.5 | +2.9 | +1.9 | +2.6 | +2.1 | +1.5 | +2.7 | +1.0 | +0.9 |
| MS (Wang et al., 2019) | 67.8 | 77.8 | 85.6 | **87.8** | 92.7 | 95.3 | 76.9 | 89.8 | 95.9 | 90.1 | 97.6 | 98.4 |
| +Metrix | **71.4** | 80.6 | 86.8 | **89.6** | **94.2** | 96.0 | 81.0 | 92.0 | **97.2** | **92.2** | **98.5** | 98.6 |
| | +3.6 | +2.8 | +1.2 | +1.8 | +1.5 | +0.7 | +4.1 | +2.2 | +1.3 | +2.1 | +0.9 | +0.2 |
| PA (Kim et al., 2020c)* | **69.7** | **80.0** | 87.0 | 87.7 | **92.9** | **95.8** | – | – | – | – | – | – |
| PA (Kim et al., 2020c) | 69.5 | 79.3 | 87.0 | 87.6 | 92.3 | 95.5 | 79.1 | 90.8 | 96.2 | 90.0 | 97.4 | 98.2 |
| +Metrix | 71.0 | **81.8** | **88.2** | 89.1 | 93.6 | **96.7** | **81.3** | 91.7 | 96.9 | 91.9 | 98.2 | **98.8** |
| | +1.3 | +1.8 | +1.2 | +1.4 | +0.7 | +0.9 | +2.2 | +0.9 | +0.7 | +1.9 | +0.8 | +0.6 |
| ProxyNCA++ (Teh et al., 2020)* | 69.0 | 79.8 | 87.3 | 86.5 | 92.5 | 95.7 | **80.7** | **92.0** | **96.7** | 90.4 | **98.1** | **98.8** |
| ProxyNCA++ (Teh et al., 2020) | 69.1 | 79.5 | **87.7** | 86.6 | 92.1 | 95.4 | 80.4 | 91.7 | **96.7** | 90.2 | 97.6 | 98.4 |
| +Metrix | 70.4 | 80.6 | 88.7 | 88.5 | 93.4 | 96.5 | **81.3** | **92.7** | 97.1 | 91.9 | 98.1 | 98.8 |
| | +1.3 | +0.8 | +1.0 | +1.9 | +0.9 | +0.8 | +0.6 | +0.7 | +0.4 | +1.5 | +0.0 | +0.0 |
| Gain over SOTA | +1.7 | +1.8 | +0.5 | +1.8 | +1.3 | +0.9 | +0.6 | +0.7 | +0.5 | +1.2 | +0.4 | +0.0 |

Table 2: *Improving the SOTA with our Metrix* (Metrix/feature) using Resnet-50 with embedding size $d = 512$. R@$K$ (%): Recall@$K$; higher is better. *: reported by authors. **Bold black**: best baseline (previous SOTA, one per column). **Red**: Our new SOTA. Gain over SOTA is over best baseline. MS: Multi-Similarity, PA: Proxy Anchor. Additional results are in subsection B.3 of the Appendix.

In terms of Recall@1, Metrix/input outperforms $i$-Mix with anchor-negative pairs by 0.5% (65.8 → 66.3) on CUB200, 0.9% (82.0 → 82.9) on Cars196, 0.6% (75.2 → 75.8) and 0.6% (87.1 → 87.7) on In-Shop. Metrix/embed outperforms MoCHI with anchor-negative pairs by 1.2% (65.2 → 66.4) on CUB200, 1.4% (82.5 → 83.9) on Cars196, 0.9% (75.8 → 76.7) and 1.2% (87.2 → 88.4) on In-Shop. The gain over MoCHi with negative-negative pairs is significantly higher. Metrix/embed also outperforms PS by 0.4% (70.0 → 70.4) on CUB200, 1% (87.9 → 88.9) on Cars196, 1% (79.6 → 80.6) on SOP and 1.3% (90.3 → 91.6) on In-Shop.

**Computational complexity**   We study the computational complexity of Metrix in subsection B.3.

**Ablation study**   We study the effect of the number $k$ of hard negatives, mixup types (input, feature and embedding), mixing pairs and mixup strength $w$ in subsection B.4 of the Appendix.

### 4.3   HOW DOES MIXUP IMPROVE REPRESENTATIONS?

We analyze how Metrix improves representation learning, given the difference between distributions at training and inference. As discussed in section 1, since the classes at inference are unseen at training, one might expect interpolation-based data augmentation like mixup to be even more important than in classification. This is so because, by mixing examples during training, we are exploring areas of the embedding space beyond the training classes. We hope that this exploration would possibly lead the model to implicitly learn a representation more appropriate for the test classes, if the distribution of the test classes lies near these areas.

**Alignment and Uniformity**   In terms of quantitative measures of properties of the training and test distributions, we follow Wang & Isola (2020). This work introduces two measures – *alignment* and *uniformity* (the lower the better) to be used both as loss functions (on the training set) and as evaluation metrics (on the test set). *Alignment* measures the expected pairwise distance between positive examples in the embedding space. A small value of alignment indicates that the positive examples are clustered together. *Uniformity* measures the (log of the) expected pairwise similarity between all examples regardless of class, using a Gaussian kernel as similarity. A small value of uniformity indicates that the distribution is more uniform over the embedding space, which is particularly relevant to our problem. Meant for contrastive learning, (Wang & Isola, 2020) use the same training and test classes, while in our case they are different.

| | | CUB200 | | | Cars196 | | | SOP | | | In-Shop | | |
|---|---|---|---|---|---|---|---|---|---|---|---|---|---|
| Method | Mixing Pairs | R@1 | R@2 | R@4 | R@1 | R@2 | R@4 | R@1 | R@10 | R@100 | R@1 | R@10 | R@20 |
| Cont (Hadsell et al., 2006) | – | 64.7 | 75.9 | 84.6 | 81.6 | 88.2 | 92.7 | 74.9 | 87.0 | 93.9 | 86.4 | 94.7 | 96.3 |
| + i-Mix (Lee et al., 2021) | anc-neg | 65.8 | 76.2 | 84.9 | 82.0 | 88.5 | 93.2 | 75.2 | 87.3 | 94.2 | 87.1 | 95.4 | 96.1 |
| + Metrix/input | pos-neg / anc-neg | 66.3 | 77.1 | 85.2 | 82.9 | 89.3 | 93.7 | 75.8 | 87.8 | 94.6 | 87.7 | 95.9 | 96.5 |
| +MoCHi (Kalantidis et al., 2020) | neg-neg | 63.1 | 74.3 | 83.8 | 76.3 | 84.0 | 89.3 | 68.9 | 83.1 | 91.8 | 81.8 | 91.9 | 93.9 |
| +MoCHi (Kalantidis et al., 2020) | anc-neg | 65.2 | 75.8 | 84.2 | 82.5 | 88.0 | 92.9 | 75.8 | 87.1 | 94.8 | 87.2 | 92.8 | 94.9 |
| +Metrix/embed | pos-neg / anc-neg | **66.4** | **77.6** | **85.4** | **83.9** | **90.3** | **94.1** | **76.7** | **88.6** | **95.2** | **88.4** | **95.4** | **96.9** |
| PA (Kim et al., 2020c) | – | 69.7 | 80.0 | 87.0 | 87.6 | 92.3 | 95.5 | 79.1 | 90.8 | 96.2 | 90.0 | 97.4 | 98.2 |
| +PS (Gu et al., 2021) | pos-neg / neg-neg | 70.0 | 79.8 | 87.2 | 87.9 | 92.8 | 95.6 | 79.6 | 90.9 | 96.4 | 90.3 | 97.4 | 98.0 |
| +Metrix/embed | pos-neg / anc-neg | **70.4** | **81.1** | **87.9** | **88.9** | **93.3** | **96.4** | **80.6** | **91.7** | **96.6** | **91.6** | **98.3** | **98.3** |

Table 3: *Comparison of our Metrix/embed with other mixing methods* using R-50 with embedding size $d = 512$. R@$K$ (%): Recall@$K$; higher is better. PA: Proxy Anchor, PS: Proxy Synthesis.

By training with contrastive loss on CUB200 and then measuring on the test set, we achieve an alignment (lower the better) of 0.28 for contrastive loss, 0.28 for $i$-Mix (Lee et al., 2021) and 0.19 for Metrix/input. MoCHi (Kalantidis et al., 2020) and Metrix/embed achieve an alignment of 0.19 and 0.17, respectively. We also obtain a uniformity (lower the better) of $-2.71$ for contrastive loss, $-2.13$ for $i$-Mix and $-3.13$ for Metrix/input. The uniformity of MoCHi and Metrix/embed is $-3.18$ and $-3.25$, respectively. This indicates that Metrix helps obtain a test distribution that is more uniform over the embedding space, where classes are better clustered and better separated.

**Utilization** The measures proposed by (Wang & Isola, 2020) are limited to a single distribution or dataset, either the training set (as loss functions) or the test set (as evaluation metrics). It is more interesting to measure the extent to which a test example, seen as a query, lies near any of the training examples, clean or mixed. For this, we introduce the measure of *utilization* $u(Q, X)$ of the training set $X$ by the test set $Q$ as

$$u(Q, X) = \frac{1}{|Q|} \sum_{q \in Q} \min_{x \in X} \|f(q) - f(x)\|^2 \tag{12}$$

Utilization measures the average, over the test set $Q$, of the minimum distance of a query $q$ to a training example $x \in X$ in the embedding space of the trained model $f$ (lower is better). A low value of utilization indicates that there are examples in the training set that are similar to test examples. When using mixup, we measure utilization as $u(Q, \hat{X})$, where $\hat{X}$ is the augmented training set including clean and mixed examples over a number of epochs and $f$ remains fixed. Because $X \subset \hat{X}$, we expect $u(Q, \hat{X}) < u(Q, X)$, that is, the embedding space is better explored in the presence of mixup.

By using contrastive loss on CUB200, utilization drops from 0.41 to 0.32 when using Metrix. This indicates that test samples are indeed closer to mixed examples than clean in the embedding space. This validates our hypothesis that a representation more appropriate for test classes is implicitly learned during exploration of the embedding space in the presence of mixup.

## 5 Conclusion

Based on the argument that metric learning is binary classification of pairs of examples into "positive" and "negative", we have introduced a direct extension of mixup from classification to metric learning. Our formulation is generic, applying to a large class of loss functions that separate positives from negatives per anchor and involve component functions that are additive over examples. Those are exactly loss functions that require less mining. We contribute a principled way of interpolating labels, such that the interpolation factor affects the relative weighting of positives and negatives. Other than that, our approach is completely agnostic with respect to the mixup method, opening the way to using more advanced mixup methods for metric learning.

We consistently outperform baselines using a number of loss functions on a number of benchmarks and we improve the state of the art using a single loss function on all benchmarks, while previous state of the art was not consistent in this respect. Surprisingly, this loss function, multi-similarity Wang et al. (2019), is not the state of the art without mixup. Because metric learning is about generalizing to unseen classes and distributions, our work may have applications to other such problems, including transfer learning, few-shot learning and continual learning.

## 6 ACKNOWLEDGEMENT

Shashanka's work was supported by the ANR-19-CE23-0028 MEERQAT project and was performed using the HPC resources from GENCI-IDRIS Grant 2021 AD011011709R1. Bill's work was partially supported by the EU RAMONES project grant No. 101017808 and was performed using the HPC resources from GRNET S.A. project pr011028. This work was partially done when Yannis was at Inria.

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

# A    MORE ON THE METHOD

## A.1    MIXED LOSS FUNCTION

**Interpretation**    To better understand the two contributions of a labeled embedding $(v, y)$ in $V(a)$ to the positive and negative terms of (9), consider the case of positive-negative mixing pairs, $M(a) \subset U^+(a) \times U^-(a)$. Then, for $((x, y), (x', y')) \in M(a)$, the mixed label is $\text{mix}_\lambda(y, y') = \text{mix}_\lambda(1, 0) = \lambda$ and (9) becomes

$$\widetilde{\ell}(a; \theta) = \tau \left( \sigma^+ \left( \sum_{(v,\lambda) \in V(a)} \lambda \rho^+(s(a, v)) \right) + \sigma^- \left( \sum_{(v,\lambda) \in V(a)} (1 - \lambda) \rho^-(s(a, v)) \right) \right). \quad (13)$$

Thus, the mixed embedding $v$ is both positive (with weight $\lambda$) and negative (with weight $1 - \lambda$). Whereas for positive-positive mixing, that is, for $M(a) \subset U^+(a)^2$, the mixed label is 1 and the negative term vanishes. Similarly, for negative-negative mixing, that is, for $M(a) \subset U^-(a)^2$, the mixed label is 0 and the positive term vanishes.

In the particular case of contrastive (1) loss, positive-negative mixing (13) becomes

$$\widetilde{\ell}_{\text{cont}}(a; \theta) := \sum_{(v,\lambda) \in V(a)} -\lambda s(a, v) + \sum_{(v,\lambda) \in V(a)} (1 - \lambda)[s(a, v) - m]_+. \quad (14)$$

Similarly, for multi-similarity (2),

$$\widetilde{\ell}_{\text{MS}}(a; \theta) := \frac{1}{\beta} \log \left( 1 + \sum_{(v,\lambda) \in V(a)} \lambda e^{-\beta(s(a,v)-m)} \right) +$$
$$\frac{1}{\gamma} \log \left( 1 + \sum_{(v,\lambda) \in V(a)} (1 - \lambda) e^{\gamma(s(a,v)-m)} \right). \quad (15)$$

## A.2    ANALYSIS: MIXED EMBEDDINGS AND POSITIVITY

**Positivity**    Under positive-negative mixing, (13) shows that a mixed embedding $v$ with interpolation factor $\lambda$ behaves as both positive and negative to different extents, depending on $\lambda$: mostly positive for $\lambda$ close to 1, mostly negative for $\lambda$ close to 0. The net effect depends on the derivative of the loss with respect to the similarity $\partial \widetilde{\ell}(a; \theta) / \partial s(a, v)$: if the derivative is negative, then $v$ behaves as positive and vice versa. This is clear from the chain rule

$$\frac{\partial \widetilde{\ell}(a; \theta)}{\partial v} = \frac{\partial \widetilde{\ell}(a; \theta)}{\partial s(a, v)} \cdot \frac{\partial s(a, v)}{\partial v}, \quad (16)$$

because $\partial s(a, v) / \partial v$ is a vector pointing in a direction that makes $a, v$ more similar and the loss is being minimized. Let $\text{Pos}(a, v)$ be the event that $v$ behaves as "positive", *i.e.*, $\partial \widetilde{\ell}(a; \theta) / \partial s(a, v) \leq 0$ and minimizing the loss will increase the similarity $s(a, v)$.

**Multi-similarity**    We estimate the probability of $\text{Pos}(a, v)$ as a function of $\lambda$ in the case of multi-similarity with a single embedding $v$ obtained by mixing a positive with a negative:

$$\widetilde{\ell}_{\text{MS}}(a; \theta) = \frac{1}{\beta} \log \left( 1 + \lambda e^{-\beta(s(a,v)-m)} \right) + \frac{1}{\gamma} \log \left( 1 + (1 - \lambda) e^{\gamma(s(a,v)-m)} \right). \quad (17)$$

In this case, $\text{Pos}(a, v)$ occurs if and only if

$$\frac{\partial \widetilde{\ell}_{\text{MS}}(a; \theta)}{\partial s(a, v)} = \frac{-\lambda e^{-\beta(s(a,v)-m)}}{(1 + \lambda e^{-\beta(s(a,v)-m)})} + \frac{(1 - \lambda) e^{\gamma(s(a,v)-m)}}{(1 + (1 - \lambda) e^{\gamma(s(a,v)-m)})} \leq 0. \quad (18)$$

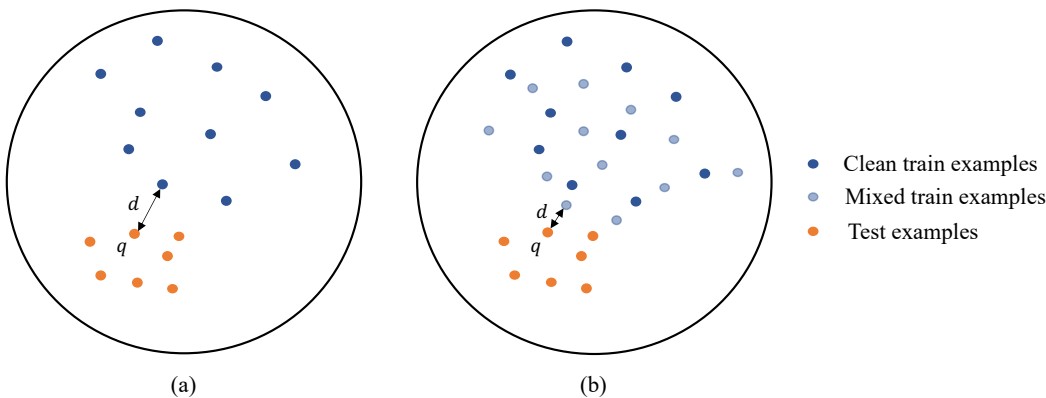

Figure 3: Exploring the embedding space when using (a) only clean examples (b) clean and mixed examples. Given a query $q$, the distance $d$ to its nearest training embedding (clean or mixed) is smaller with mixup (b) than without (a).

By letting $t := s(a, v) - m$, this condition is equivalent to

$$\frac{(1 - \lambda)e^{\gamma t}}{(1 + (1 - \lambda)e^{\gamma t})} \leq \frac{\lambda e^{-\beta t}}{(1 + \lambda e^{-\beta t})} \tag{19}$$

$$(1 - \lambda)e^{\gamma t}(1 + \lambda e^{-\beta t}) \leq \lambda e^{-\beta t}(1 + (1 - \lambda)e^{\gamma t}) \tag{20}$$

$$(1 - \lambda)e^{\gamma t} + \lambda(1 - \lambda)e^{(\gamma - \beta)t} \leq \lambda e^{-\beta t} + \lambda(1 - \lambda)e^{(\gamma - \beta)t} \tag{21}$$

$$e^{(\beta + \gamma)t} \leq \frac{\lambda}{1 - \lambda} \tag{22}$$

$$(\beta + \gamma)(s(a, v) - m) \leq \ln\left(\frac{\lambda}{1 - \lambda}\right) \tag{23}$$

$$s(a, v) \leq \frac{1}{\beta + \gamma} \ln\left(\frac{\lambda}{1 - \lambda}\right) + m. \tag{24}$$

Finally, the probability of $\text{Pos}(a, v)$ as a function of $\lambda$ is

$$\text{P}(\text{Pos}(a, v)) = F_\lambda\left(\frac{1}{\beta + \gamma} \ln\left(\frac{\lambda}{1 - \lambda}\right) + m\right), \tag{25}$$

where $F_\lambda$ is the CDF of similarities $s(a, v)$ between anchors $a$ and mixed embeddings $v$ with interpolation factor $\lambda$.

In Figure 2, we measure the probability of $\text{Pos}(a, v)$ as a function of $\lambda$ in two ways. First, we measure the derivative $\partial\widetilde{\ell}_{\text{MS}}(a; \theta)/\partial s(a, v)$ for anchors $a$ and mixed embeddings $v$ over the entire dataset and we report the empirical probability of this derivative being non-positive versus $\lambda$. Second, we measure $\text{P}(\text{Pos}(a, v))$ theoretically using (25), where the CDF of similarities $s(a, v)$ is again measured empirically for $a$ and $v$ over the dataset, as a function of $\lambda$. Despite the simplifying assumption of a single positive and a single negative in deriving (25), we observe that the two measurements agree in general. They are both increasing functions of $\lambda$ of sigmoidal shape, they roughly yield $\text{P}(\text{Pos}(a, v)) \geq 0.5$ for $\lambda \geq 0.5$ and they confirm that a mixed embedding is mostly positive for $\lambda$ close to 1 and mostly negative for $\lambda$ close to 0.

### A.3 MORE ON UTILIZATION

In subsection 4.3, we discuss that a representation more appropriate for test classes is implicitly learned during exploration of the embedding space in the presence of mixup. We provide an illustration of this exploration in Figure 3, where we visualize the embedding space using (a) only clean train examples and (b) clean and mixed train examples. In case (a), the model is trained using only clean examples, exploring a smaller area of the embedding space. In case (b), it is trained using both

| DATASET | CUB200 (Wah et al., 2011) | CARS196 (Krause et al., 2013) | SOP (Oh Song et al., 2016) | IN-SHOP (Liu et al., 2016) |
|---|---|---|---|---|
| Objects | birds | cars | household furniture | clothes |
| # classes | 200 | 196 | 22, 634 | 7, 982 |
| # training images | 5, 894 | 8, 092 | 60, 026 | 26, 356 |
| # testing images | 5, 894 | 8, 093 | 60, 027 | 26, 356 |
| # training classes | 100 | 98 | 11, 318 | 3991 |
| # testing classes | 100 | 98 | 11, 318 | 3991 |
| sampling | random | random | balanced | balanced |
| samples per class | – | – | 5 | 5 |
| classes per batch | $65^{\dagger}$ | $70^{\dagger}$ | 20 | 20 |
| learning rate | $1 \times 10^{-4}$ | $1 \times 10^{-4}$ | $3 \times 10^{-5}$ | $1 \times 10^{-4}$ |

Table 4: *Statistics and settings* for the four datasets we use in our experiments. $^{\dagger}$: average.

mixed and clean examples, exploring a larger area. It is clear that the distance between a query and its nearest training example (clean or mixup) is smaller in the presence of mixup. Utilization is the average of this distance over the test set. This shows that the model implicitly learns a representation closer the test example in the presence of mixup during training and it partially explains why mixup leads to better performance.

## B    MORE ON EXPERIMENTS

### B.1    SETUP

**Datasets and sampling**    Dataset statistics are summarized in Table 4. Since the number of classes is large compared to the batch size in SOP and In-Shop, batches would rarely contain a positive pair when sampled uniformly at random. Hence, we use *balanced sampling* (Zhai & Wu, 2018), *i.e.*, a fixed number of classes and examples per class, as shown in Table 4. For fair comparison with baseline methods, images are randomly flipped and cropped to $224 \times 224$ at training. At inference, we resize to $256 \times 256$ and then center-crop to $224 \times 224$.

**Training**    We train R-50 using AdamW (Loshchilov & Hutter, 2019) optimizer for 100 epochs with a batch size 100. The initial learning rate per dataset is shown in Table 4. The learning rate is decayed by 0.1 for Cont and by 0.5 for MS and PA on CUB200 and Cars196. For SOP and In-Shop, we decay the learning rate by 0.25 for all losses. The weight decay is set to 0.0001.

### B.2    MIXUP SETTINGS

In mixup for classification, given a batch of $n$ examples, it is standard to form $n$ pairs of examples by pairing the batch with a *random permutation* of itself, resulting in $n$ mixed examples, either for input or manifold mixup. In metric learning, it is common to obtain $n$ embeddings and then use all $\frac{1}{2}n(n-1)$ pairs of embeddings in computing the loss. We thus treat mixup types differently.

**Input mixup**    Mixing all pairs would be computationally expensive in this case, because we would compute $\frac{1}{2}n(n-1)$ embeddings. A random permutation would not produce as many hard examples as can be found in all pairs. Thus, for each anchor (each example in the batch), we use the $k$ *hardest negative* examples and mix them with positives or with the anchor. We use $k = 3$ by default.

**Manifold mixup**    Originally, manifold mixup (Verma et al., 2019) focuses on the *first* few layers of the network. Mixing all pairs would then be even more expensive than input mixup, because intermediate features (tensors) are even larger than input examples. Hence, we focus on the *last* few layers instead, where features and embeddings are compact, and we mix all pairs. We use feature mixup by default and call it *Metrix/feature* or just *Metrix*, while input and embedding mixup are called *Metrix/input* and *Metrix/embed*, respectively. All options are studied in subsection B.4.

**Mixing pairs**    Whatever the mixup type, we use clean examples as anchors and we define a set $M(a)$ of pairs of examples to mix for each anchor $a$, with their labels (positive or negative). By default, we mix positive-negative or anchor-negative pairs, according to $M(a) := U^{+}(a) \times U^{-}(a)$

| | CUB200 | | | CARS196 | | | SOP | | | IN-SHOP | | |
|---|---|---|---|---|---|---|---|---|---|---|---|---|
| Method | 1 | 2 | 4 | 1 | 2 | 4 | 1 | 10 | 100 | 1 | 10 | 20 |
| Triplet (Weinberger & Saul, 2009) | 63.5 | 75.6 | 84.4 | 77.3 | 85.4 | 90.8 | 70.5 | 85.6 | 94.3 | 85.3 | 96.6 | 97.8 |
| LiftedStructure (Oh Song et al., 2016) | 65.9 | 75.8 | 84.5 | 81.4 | 88.3 | 92.4 | 76.1 | 88.6 | 95.2 | 88.6 | 97.6 | 98.4 |
| ProxyNCA (Movshovitz-Attias et al., 2017) | 65.2 | 75.6 | 83.8 | 81.2 | 87.9 | 92.6 | 73.2 | 87.0 | 94.4 | 86.2 | 95.9 | 97.0 |
| Margin (Wu et al., 2017) | 65.0 | 76.2 | 84.6 | 82.1 | 88.7 | 92.7 | 74.8 | 87.8 | 94.8 | 88.6 | 97.0 | 97.8 |
| SoftTriple (Qian et al., 2019) | 67.3 | 77.7 | 86.2 | 86.5 | 91.9 | 95.3 | 79.8 | 91.2 | 96.3 | **91.0** | 97.6 | 98.3 |
| D&C (Sanakoyeu et al., 2019)[*] | 65.9 | 76.6 | 84.4 | 84.6 | 90.7 | 94.1 | 75.9 | 88.4 | 94.9 | 85.7 | 95.5 | 96.9 |
| EPSHN (Xuan et al., 2020)[*] | 64.9 | 75.3 | 83.5 | 82.7 | 89.3 | 93.0 | 78.3 | 90.7 | 96.3 | 87.8 | 95.7 | 96.8 |
| ProxyNCA++ (Teh et al., 2020)[*] | 69.0 | 79.8 | **87.3** | 86.5 | 92.5 | 95.7 | **80.7** | 92.0 | 96.7 | 90.4 | **98.1** | **98.8** |
| Cont (Hadsell et al., 2006) | 64.7 | 75.9 | 84.6 | 81.6 | 88.2 | 92.7 | 74.9 | 87.0 | 93.9 | 86.4 | 94.7 | 96.2 |
| +Metrix/input | 66.3 | 77.1 | 85.2 | 82.9 | 89.3 | 93.7 | 75.8 | 87.8 | 94.6 | 87.7 | 95.9 | 96.5 |
| | +1.6 | +1.2 | +0.6 | +1.3 | +1.1 | +1.0 | +0.9 | +0.8 | +0.7 | +1.3 | +1.2 | +0.3 |
| +Metrix | 67.4 | 77.9 | 85.7 | 85.1 | 91.1 | 94.6 | 77.5 | 89.1 | 95.5 | 89.1 | 95.7 | 97.1 |
| | +2.7 | +2.0 | +1.1 | +3.5 | +2.9 | +1.9 | +2.6 | +2.1 | +1.5 | +2.7 | +1.0 | +0.9 |
| +Metrix/embed | 66.4 | 77.6 | 85.4 | 83.9 | 90.3 | 94.1 | 76.7 | 88.6 | 95.2 | 88.4 | 95.4 | 96.8 |
| | +1.7 | +1.7 | +0.8 | +2.3 | +2.1 | +1.4 | +1.8 | +1.6 | +1.3 | +2.0 | +0.7 | +0.6 |
| MS (Wang et al., 2019) | 67.8 | 77.8 | 85.6 | **87.8** | 92.7 | 95.3 | 76.9 | 89.8 | 95.9 | 90.1 | 97.6 | 98.4 |
| +Metrix/input | 69.0 | 79.1 | 86.0 | 89.0 | 93.4 | 96.0 | 77.9 | 90.6 | 95.9 | 91.8 | 98.0 | 98.9 |
| | +1.2 | +1.3 | +0.4 | +1.2 | +0.7 | +0.7 | +1.0 | +0.8 | +0.0 | +1.7 | +0.4 | +0.5 |
| +Metrix | **71.4** | 80.6 | 86.8 | **89.6** | **94.2** | 96.0 | 81.0 | 92.0 | **97.2** | **92.2** | **98.5** | 98.6 |
| | +3.6 | +2.8 | +1.2 | +1.8 | +1.5 | +0.7 | +4.1 | +2.2 | +1.3 | +2.1 | +0.9 | +0.2 |
| +Metrix/embed | 70.2 | 80.4 | 86.7 | 88.8 | 92.9 | 95.6 | 78.5 | 91.3 | 96.7 | 91.9 | 98.3 | 98.7 |
| | +2.4 | +2.6 | +1.1 | +1.0 | +0.2 | +0.3 | +1.6 | +1.5 | +0.8 | +1.8 | +0.7 | +0.3 |
| PA (Kim et al., 2020c)[*] | **69.7** | **80.0** | 87.0 | 87.7 | **92.9** | 95.8 | – | – | – | – | – | – |
| PA (Kim et al., 2020c) | 69.5 | 79.3 | 87.0 | 87.6 | 92.3 | 95.5 | 79.1 | 90.8 | 96.2 | 90.0 | 97.4 | 98.2 |
| +Metrix/input | 70.5 | 81.2 | 87.8 | 88.2 | 93.2 | 96.2 | 79.8 | 91.4 | 96.5 | 90.9 | 98.1 | 98.4 |
| | +0.8 | +1.2 | +0.8 | +0.5 | +0.3 | +0.4 | +0.7 | +0.6 | +0.3 | +0.9 | +0.7 | +0.2 |
| +Metrix | 71.0 | **81.8** | **88.2** | 89.1 | 93.6 | **96.7** | **81.3** | 91.7 | 96.9 | 91.9 | 98.2 | **98.8** |
| | +1.3 | +1.8 | +1.2 | +1.4 | +0.7 | +0.9 | +2.2 | +0.9 | +0.7 | +1.9 | +0.8 | +0.6 |
| +Metrix/embed | 70.4 | 81.1 | 87.9 | 88.9 | 93.3 | 96.4 | 80.6 | 91.7 | 96.6 | 91.6 | 98.3 | 98.3 |
| | +0.7 | +1.1 | +0.9 | +1.2 | +0.4 | +0.6 | +1.5 | +0.9 | +0.4 | +1.6 | +0.9 | +0.1 |
| ProxyNCA++ (Teh et al., 2020)[*] | 69.0 | 79.8 | 87.3 | 86.5 | 92.5 | 95.7 | **80.7** | 92.0 | 96.7 | 90.4 | **98.1** | **98.8** |
| ProxyNCA++ (Teh et al., 2020) | 69.1 | 79.5 | **87.7** | 86.6 | 92.1 | 95.4 | 80.4 | 91.7 | **96.7** | 90.2 | 97.6 | 98.4 |
| +Metrix/input | 69.7 | 79.9 | 88.3 | 87.5 | 92.9 | 96.0 | 80.9 | 92.2 | 96.9 | 91.4 | 98.1 | 98.8 |
| | +0.6 | +0.1 | +0.6 | +0.9 | +0.4 | +0.3 | +0.2 | +0.2 | +0.2 | +1.0 | +0.0 | +0.0 |
| +Metrix | 70.4 | 80.6 | 88.7 | 88.5 | 93.4 | 96.5 | **81.3** | **92.7** | 97.1 | 91.9 | 98.1 | 98.8 |
| | +1.3 | +0.8 | +1.0 | +1.9 | +0.9 | +0.8 | +0.6 | +0.7 | +0.4 | +1.5 | +0.0 | +0.0 |
| +Metrix/ embed | 70.2 | 80.2 | 88.2 | 88.1 | 93.0 | 96.2 | 81.1 | 92.4 | 97.0 | 91.6 | 98.1 | 98.8 |
| | +1.1 | +0.4 | +0.5 | +1.5 | +0.5 | +0.5 | +0.4 | +0.4 | +0.3 | +1.2 | +0.0 | +0.0 |
| Gain over SOTA | +1.7 | +1.8 | +0.5 | +1.8 | +1.3 | +0.9 | +0.6 | +0.0 | +0.5 | +1.2 | +0.4 | +0.0 |

Table 5: *Improving the SOTA with our Metrix* (Metrix/feature) using Resnet-50 with embedding size $d = 512$. R@$K$ (%): Recall@$K$; higher is better. [*]: reported by authors. **Bold black**: best baseline (previous SOTA, one per column). **Red**: Our new SOTA. Gain over SOTA is over best baseline. MS: Multi-Similarity, PA: Proxy Anchor

and $M(a) := S(a) \times U^-(a)$, respectively, where $U^-(a)$ is replaced by hard negatives only for input mixup. The two options are combined by choosing uniformly at random in each iteration. More options are studied in subsection B.4.

**Hyper-parameters** For any given mixup type or set of mixup pairs, the interpolation factor $\lambda$ is drawn from Beta$(\alpha, \alpha)$ with $\alpha = 2$. We empirically set the mixup strength (10) to $w = 0.4$ for positive-negative pairs and anchor-negative pairs.

## B.3 MORE RESULTS

**Computational complexity** On CUB200 dataset, using a batch size of 100 on an NVIDIA RTX 2080 Ti GPU, the average training time in ms/batch is 586 for MS and 817 for MS+Metrix. The 39% increase in complexity is reasonable for 3.6% increase in R@1. Furthermore, the average training time in ms/batch is 483 for baseline PA, 965 for PA+Metrix and 1563 for PS (Gu et al., 2021). While the computation cost of PS is higher than Metrix by 62%, Metrix outperform PS by 0.4% and 1.3% in terms of R@1 and R@2 respectively (Table 3). At inference, the computational cost is equal for all methods.

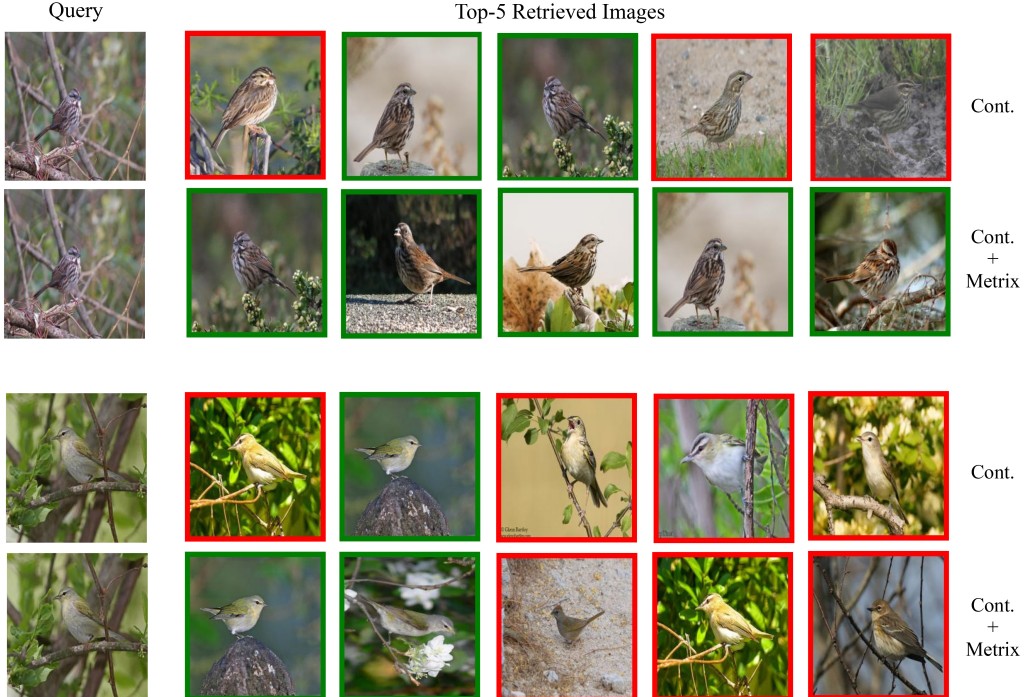

Figure 4: *Retrieval results* on CUB200 using Contrastive loss, with and without mixup. For each query, the top-5 retrieved images are shown. Images highlighted in green (red) are correctly (incorrectly) retrieved images.

**Improving the state of the art**    Table 5 is an extension of Table 2 that includes all three mixup types (input, feature, embedding). It shows that not just feature mixup but *all* mixup types consistently improve the performance of all baseline losses (Cont, MS, PA, ProxyNCA++) across all datasets. It also shows that across all baseline losses and all datasets, feature mixup works best, followed by embedding and input mixup. This result confirms the findings of Table 6 on Cars196.

**Qualitative results of retrieval**    Figure 4 shows qualitative results of retrieval on CUB200 using Contrastive loss, with and without mixup. This dataset has large intra-class variations such as pose variation and background clutter. Baseline Contrastive loss may fail to retrieve the correct images due to these challenges. The ranking is improved in the presence of mixup.

**Visualization of embedding space**    We visualize CUB200 test examples for 10, 15 and 20 classes in the embedding space using Contrastive loss, with and without mixup in Figure 5. We observe that in the presence of mixup, the embeddings are more tightly clustered and more uniformly spread, despite the variations in pose and background in the test set. This finding validates our quantitative analysis of alignment and uniformity in subsection 4.3.

### B.4    ABLATIONS

We perform ablations on Cars196 using R-50 with $d = 512$, applying mixup on contrastive loss.

**Hard negatives**    We study the effect of the number $k$ of hard negatives using different mixup types. The set of mixing pairs is chosen from (positive-negative, anchor-negative) uniformly at random per iteration. We choose $k = 3$ for input mixup. For feature/embedding mixup, we mix all pairs in a batch by default, but also study $k \in \{20, 40\}$. As shown in Table 6, $k = 3$ for input and all pairs for feature/embedding mixup works best. Still, using few hard negatives for feature/embedding mixup is on par or outperforms input mixup. All choices significantly outperform the baseline.

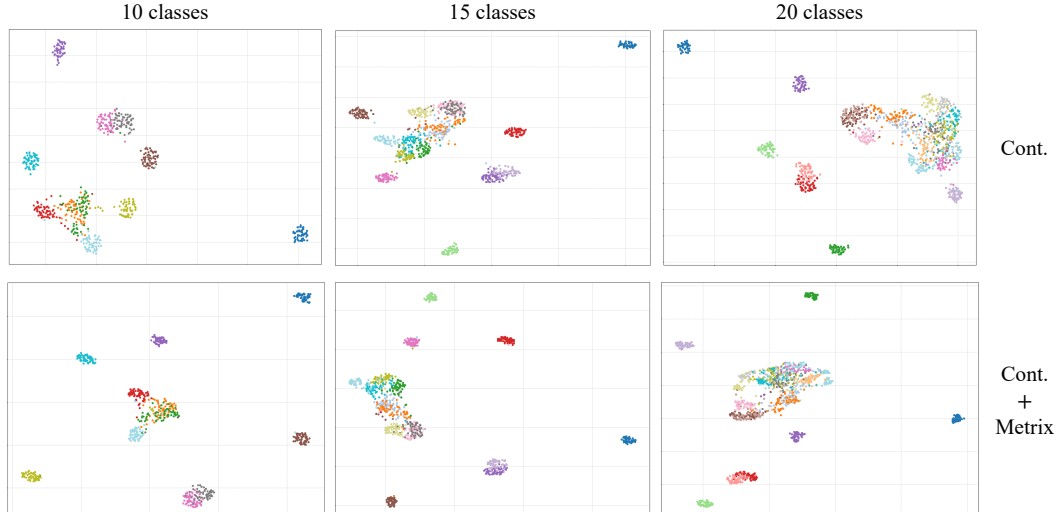

Figure 5: *Embedding space visualization* of CUB200 test examples of a given number of classes using Contrastive loss, with and without mixup.

**Mixing pairs**   We study the effect of mixing pairs $M(a)$, in particular, $U^+(a)^2$ (positive-positive), $U^+(a) \times U^-(a)$ (positive-negative) and $S(a) \times U^-(a)$ (anchor-negative), again using different mixup types. As shown in Table 6, when using a single set of mixing pairs during training, positive-negative and anchor-negative consistently outperform the baseline, while positive-positive is actually outperformed by the baseline. This may be due to the lack of negatives in the mixed loss (9), despite the presence of negatives in the clean loss (3). Hence, we only use positive-negative and anchor-negative by default, combined by choosing uniformly at random in each iteration.

**Mixup types**   We study the effect of mixup type (input, feature, embedding), when used alone. The set of mixing pairs is chosen from (positive-negative, anchor-negative) uniformly at random per iteration. As shown in both "hard negatives" and "mixing pairs" parts of Table 6, our default feature mixup works best, followed by embedding and input mixup.

**Mixup type combinations**   We study the effect of using more than one mixup type (input, feature, embedding), chosen uniformly at random per iteration. The set of mixing pairs is also chosen from (positive-negative, anchor-negative) uniformly at random per iteration. As shown in Table 6, mixing inputs, features and embeddings works best. Although this solution outperforms feature mixup alone by $0.2\%$ Recall@1 ($85.1 \rightarrow 85.3$), it is computationally expensive because of using input mixup. The next best efficient choice is mixing features and embeddings, which however is worse than mixing features alone ($84.7$ *vs.* $85.1$). This is why we chose feature mixup by default.

**Mixup strength** $w$   We study the effect of the mixup strength $w$ in the combination of the clean and mixed loss (10) for different mixup types. As shown in Figure 6, mixup consistently improves the baseline and the effect of $w$ is small, especially for input and embedding mixup. Feature mixup works best and is slightly more sensitive.

**Ablation on CUB200**   We perform additional ablations on CUB200 using R-50 with $d = 128$ by applying contrastive loss. All results are shown in Table 7. One may draw the same conclusions as from Table 6 on Cars196 with $d = 512$, which confirms that our choice of hard negatives and mixup pairs is generalizable across different datasets and embedding sizes.

In particular, following the settings of subsection B.4, we observe in Table 7 that using $k = 3$ hard negatives for input mixup and all pairs for feature/embedding mixup achieves the best performance in terms of Recall@1. Similarly, using a single set of mixing pairs, positive-negative and anchor-negative consistently outperform the baseline, whereas positive-positive is inferior than the baseline. Furthermore, combining positive-negative and anchor-negative pairs by choosing uniformly at random in each iteration achieves the best overall performance.

| STUDY | HARD NEGATIVES $k$ | MIXING PAIRS | MIXUP TYPE | R@1 | R@2 | R@4 | R@8 |
|---|---|---|---|---|---|---|---|
| baseline | | | | 81.6 | 88.2 | 92.7 | 95.8 |
| | 1 | pos-neg / anc-neg | input | 82.0 | 89.1 | 93.1 | 96.1 |
| | 2 | pos-neg / anc-neg | input | 82.5 | 89.2 | 93.4 | 96.2 |
| | 3 | pos-neg / anc-neg | input | 82.9 | 89.3 | 93.7 | 95.5 |
| hard negatives | 20 | pos-neg / anc-neg | feature | 83.5 | 90.1 | 94.0 | 96.5 |
| | 40 | pos-neg / anc-neg | feature | 84.0 | 90.4 | 94.2 | 96.8 |
| | all | pos-neg / anc-neg | feature | **85.1** | **91.1** | **94.6** | **97.0** |
| | 20 | pos-neg / anc-neg | embed | 82.7 | 89.2 | 93.4 | 96.1 |
| | 40 | pos-neg / anc-neg | embed | 83.0 | 90.0 | 93.8 | 96.4 |
| | all | pos-neg / anc-neg | embed | 83.4 | 89.9 | 94.1 | 96.4 |
| | – | pos-pos | input | 81.0 | 88.2 | 92.6 | 95.6 |
| | 3 | pos-neg | input | 82.4 | 89.1 | 93.3 | 95.6 |
| | 3 | anc-neg | input | 81.8 | 89.0 | 93.6 | 95.4 |
| mixing pairs | – | pos-pos | feature | 81.1 | 88.3 | 92.9 | 95.8 |
| | all | pos-neg | feature | 84.0 | 90.2 | 94.2 | 96.6 |
| | all | anc-neg | feature | 83.7 | 90.1 | 94.4 | 96.7 |
| | – | pos-pos | embed | 78.3 | 85.7 | 90.8 | 94.4 |
| | all | pos-neg | embed | 83.1 | 90.0 | 93.9 | 96.6 |
| | all | anc-neg | embed | 82.7 | 89.5 | 93.5 | 96.3 |
| mixup type combinations | {1, all} | pos-neg / anc-neg | {input, feature} | 83.7 | 94.2 | 95.9 | 96.7 |
| | {3, all} | pos-neg / anc-neg | {input, embed} | 83.0 | 90.9 | 94.1 | 96.4 |
| | {all, all} | pos-neg / anc-neg | {feature, embed} | 84.7 | 90.6 | 94.4 | 96.9 |
| | {1, all, all} | pos-neg / anc-neg | {input, feature, embed} | **85.3** | **94.9** | **96.2** | **97.1** |

Table 6: *Ablation study of our Metrix* using contrastive loss and R-50 with embedding size $d = 512$ on Cars196. R@$K$ (%): Recall@$K$; higher is better.

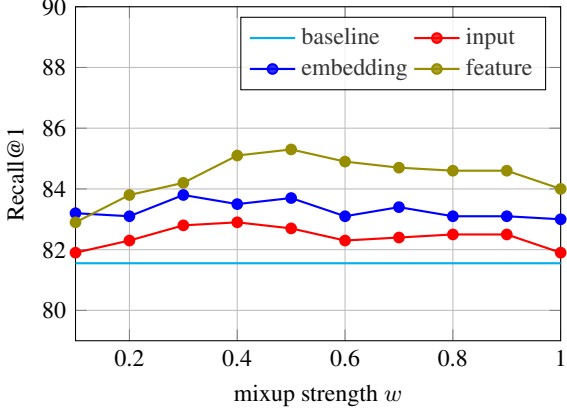

Figure 6: *Effect of mixup strength* for different mixup types using contrastive loss and R-50 with embedding size $d = 512$ on Cars196. Recall@$K$ (%): higher is better.

| STUDY | HARD NEGATIVES $k$ | MIXING PAIRS | MIXUP TYPE | R@1 | R@2 | R@4 | R@8 |
|---|---|---|---|---|---|---|---|
| baseline | | | | 61.6 | 73.7 | 83.6 | 90.1 |
| hard negatives | 1 | pos-neg / anc-neg | input | 62.4 | 73.9 | 83.0 | 89.7 |
| | 2 | pos-neg / anc-neg | input | 62.7 | 74.2 | **83.6** | 90.0 |
| | 3 | pos-neg / anc-neg | input | 63.1 | 74.5 | 83.5 | 90.3 |
| | 20 | pos-neg / anc-neg | feature | 63.9 | 75.0 | 83.9 | 89.9 |
| | 40 | pos-neg / anc-neg | feature | 63.5 | 75.2 | 83.5 | 89.8 |
| | all | pos-neg / anc-neg | feature | **64.5** | **75.4** | 84.3 | **90.6** |
| | 20 | pos-neg / anc-neg | embed | 63.1 | 74.3 | 83.1 | 90.0 |
| | 40 | pos-neg / anc-neg | embed | 63.5 | 74.7 | 83.6 | 90.1 |
| | all | pos-neg / anc-neg | embed | 64.0 | 75.1 | 84.8 | 90.9 |
| mixing pairs | – | pos-pos | input | 58.7 | 70.7 | 80.1 | 87.1 |
| | 3 | pos-neg | input | 62.9 | 75.1 | 83.4 | 90.6 |
| | 3 | anc-neg | input | 62.8 | 74.7 | 83.6 | 90.1 |
| | – | pos-pos | feature | 61.0 | 73.1 | 82.5 | 89.7 |
| | all | pos-neg | feature | 63.9 | 75.0 | 83.9 | 89.9 |
| | all | anc-neg | feature | 63.8 | 74.8 | 83.6 | 90.2 |
| | – | pos-pos | embed | 59.7 | 72.2 | 82.7 | 89.5 |
| | all | pos-neg | embed | 63.8 | 75.1 | 83.3 | 90.5 |
| | all | anc-neg | embed | 63.5 | 75.0 | 83.9 | 90.5 |
| mixup type combinations | {1, all} | pos-neg / anc-neg | {input, feature} | 63.9 | 75.1 | **84.9** | 90.5 |
| | {3, all} | pos-neg / anc-neg | {input, embed} | 63.4 | 74.9 | 84.5 | 90.1 |
| | {all, all} | pos-neg / anc-neg | {feature, embed} | 64.2 | 75.2 | 84.1 | 90.7 |
| | {1, all, all} | pos-neg / anc-neg | {input, feature, embed} | **65.3** | **76.2** | 84.4 | **91.2** |

Table 7: *Ablation study of our Metrix* using contrastive loss and R-50 with embedding size $d = 128$ on CUB200. R@$K$ (%): Recall@$K$; higher is better.

We also study the effect of using more than one mixup type (input, feature,embedding), chosen uniformly at random per iteration. The set of mixing pairs is also chosen from (positive-negative, anchor-negative) uniformly at random per iteration in this study. From Table 7, we observe that although mixing input, features and embedding works best with an improvement of $0.8\%$ over feature mixup alone ($64.5 \rightarrow 65.3$), it is computationally expensive due to using input mixup. The next best choice is mixing features and embeddings, which is worse than using feature mixup alone ($64.2$ *vs.* $64.5$). This confirms our choice of using feature mixup as default.

