# OpenReview forum: "It Takes Two to Tango: Mixup for Deep Metric Learning"
_ICLR.cc/2022/Conference — ICLR 2022 Poster_

### Official Review · Reviewer_c8hM · 2021-11-02

**Correctness:** 4
**Technical Novelty And Significance:** 2
**Empirical Novelty And Significance:** 3
**Recommendation:** 6
**Confidence:** 3

**Main Review:**

Pros.
1) It adapts a generalized formulation of loss function to the mixup technique, such that both examples (inputs) and target labels (outputs) can be integrated to yield better embeddings or representations.
2) It proposes a new metric named utilization to examine the improvements of representation in the embedding space.
3) It provides convincing empirical results on benchmark data sets.


Cons.
1) Previous works have shown that the effectiveness of the mixup for interpolating embeddings and labels (Zhang et al., 2018; Verma et al., 2019). This work borrows the same idea on input samples and labels rather than using embeddings. Such exploration should be encouraged, but the novelty is a bit weak. Other mixup variants might be better choices to improve novelty.
2) More analysis is required on error function in Eq. (10).
3) Some qualitative comparison results can be shown to further verify the improvement brought by mixup.

**Summary Of The Paper:**

This work explores the way of mixing both examples and target labels for deep metric learning. It considers metric learning loss and data augmentation technique (e.g., mixup) together when handling two or more examples at a time. A generalized formulation of loss function was modified to accommodate for the mixup technique. Also, a new metric called utilization is introduced for evaluating representation improvements. Some theoretical analysis are provided. A number of experiments were conducted on four benchmarks to show the superiority of the proposed method.

**Summary Of The Review:**

This paper is generally well written and easy to read. It provides sound technique details with theoretical analysis. Also it presents lots of experimental results to support its claim that the mixup of inputs and outputs improves the representation quality. However, the novelty is of a bit limited. Overall, it is an acceptable paper that will attract attention from relevant researchers.

---

> ### Author Response · Authors · 2021-11-18
> **Response to Reviewer c8hM**
>
> We appreciate the Reviewer c8hM's valuable feedback. We address the concerns as follows:
>
> **[Only input mixup]**
>
> We not only interpolate input, but also interpolate features and embeddings along with the targets labels, and show that it improves the performance on both pair-based and proxy-based loss functions (Table 2, Table 5).
>
> Other mixup variants such as CutMix (Yun et al., 2019), PuzzleMix (Kim et al., 2020), CoMix (Kim et al., 2021) etc. perform mixup in the image space. Applying these methods to deep metric learning  would be challenging  since computing the embeddings from all pairs in the image space would be computationally very expensive (Section B.2). To reduce computational complexity, we would need to resort to mining hard negatives whose performance is worse when compared to feature mixup where all pairs can be used (Table 5).
>
> Our key contributions do not lie in the mixup types (input/embedding/feature) or in which layer mixup is performed. We propose a generic way of representing and interpolating labels, which allows straightforward extension of different kinds of mixup to deep metric learning for a large class of loss functions. Our key contributions are enumerated at the end of Section 1.
>
> Reviewer  S9kx acknowledges that *this is the first time mixup has been applied to deep metric learning* and Reviewer gPed acknowledges that *this is good paper which has technical novelty.*
>
> **[Analysis - Equation 10]**
>
> We appreciate the reviewer's question. We address this concern in a comment to Reviewer EXEY. In our initial experiments, we trained Metrix without the clean loss in eq. 10. We observe that on CUB200 using contrastive loss with embedding size $d=512$, R@1 drops from 64.7 (clean loss only) to 54.4. By using both clean and mixed loss, the performance improves from 64.7 to 67.4.
>
> We hypothesize that by removing the clean loss in eq. 10, we loose additional information which the network requires during learning. Here, we use mixup for data augmentation and not to replace the original data.
>
> We will add this experiment and discussion in the paper.
>
> **[Qualitative results]**
>
> We will include tsne plots and some images of the retrieval in the Appendix

---

> > ### Author Response · Authors · 2021-11-22
> > **Update on Qualitative results**
> >
> > Concerning the reviewer's original comment, we also updated Section B.3 in the Appendix with some qualitative results of the retrieval and the visualization of the embedding space on CUB200.

---

> > ### Comment · Reviewer_c8hM · 2021-11-29
> > **Follow-up Response to Reviewer c8hM**
> >
> > Authors have addressed most of my concerns, and necessary revisions are desired in the later version.

---

### Official Review · Reviewer_S9kx · 2021-11-04

**Correctness:** 3
**Technical Novelty And Significance:** 2
**Empirical Novelty And Significance:** 2
**Recommendation:** 6
**Confidence:** 4

**Main Review:**

Strengths:
------------
1) A general form of loss function for DML that can be easily converted to specific losses like Contrastive, Proxy-anchor, Multi-similarity etc. + incorporating mixup to this.
2) Good experimental validation showing the effectiveness of the proposed approach compared to the baseline and the SOTA

Weaknesses:
-----------------
1) Limited contribution/significance: The two main contributions of the paper are representing the loss function in a general form and incorporating mixup into this loss function (using continuous labels instead of discrete labels as used in the existing dml). I think the idea of using a general dml loss function is inspired from Wang et. al 2019 which uses Multi-similarity loss as a general dml loss function. The loss function in Eqn. (3) is a modified version of MS loss Eqn. (2). Mixup has been extensively used in classification tasks and three different types (input/feature/embedding) have also been proposed previously. Although this is the first time mixup has been applied to dml, the contribution of the paper seems to be limited.

2) Limited scope: The proposed mixup approach is tailored to the dml problem in this work. Hence, its scope beyond dml is limitted. Although the paper mentions that the proposed approach can be extended to transfer/few-shot/continual learning, "our work may have applications to other such problems, including transfer learning, few-shot learning and continual learning", it would be difficult because the method assumes a particular form of loss function which may not hold true in other cases.

3) Experiments: It would interesting to see some images for the following cases:
a) the mixup training (for input mixup) images for different values of \lambda.
b) how the tsne plots for the mixup samples would look like?
c) the nearest neighbors for different values of \lambda (obtained by comparing the features in the embedding space)



**Summary Of The Paper:**

The paper presents a technique for using mixup augmentation in deep metric learning training. Specifically, the dml loss function is represented in a general form so that mixup loss can be easily computed for different pairs. This loss is combined with regular dml loss for training the network. The method is evaluated on CUB200, CARS196, SOP and IN-SHOP datasets where it outperforms it's baselines and also achieves SOTA results.

**Summary Of The Review:**

Although the proposed approach seems to be novel (applying mixup to dml), the scope of the technique is limited to the dml applications only and the contributions are marginal. Also, showing some images of the retrieval, tsne plots would be helpful and can provide more insights into the problem.

---

> ### Author Response · Authors · 2021-11-18
> **Response to Reviewer S9kx**
>
> We appreciate the Reviewer S9kx's valuable feedback. We address the concerns as follows:
>
> **[Significance]**
>
> The idea of using a generic deep metric loss function is inspired from MS (Wang et al., 2019), PA (Kim et al., 2020) and SPCE (Boudiaf et al., 2020). Although MS and our work propose generic loss functions for deep metric learning, the motivation is different. The generic loss function in MS is based on pairwise weighting, which is expressed as a function where the weights of the positive and negative pairs are not binary. Through this generic formulation, they encourage researchers to develop new loss functions for metric learning that incorporate non-binary weights of positive and negative pairs.
>
> In this work, our motivation of the generic loss function is different. We do not propose a new loss function for metric learning. Our generic loss function encompasses several different metric learning losses (eq. 3, Table 1) such that it can be easily adapted in the presence of mixup. The generic formulation of MS is expressed as the derivative of loss with respect to the pairs that define the weights and not the loss function itself. Thus, it would not help towards adapting it for mixup.
>
> As the reviewer acknowledges that *this is the first time mixup has been applied to deep metric learning*, we believe that this in itself is an important contribution.  Reviewer gPed also acknowledges that *this is good paper which has technical novelty, where the generalized loss for metrics learning allows to incorporate mixup augmentation*
>
> **[Limited scope]**
>
> Incremental learning work such as [1*] uses triplet loss where the positive and negative pairs are obtained as embeddings from two different networks. Few-shot learning works [2*, 3*] use modified NCA loss. All these works across different domains use deep metric learning losses. This validates our claim that our work may have applications to other such problems, including few-shot learning, continual/incremental learning. Thus, our proposed mixup methods can be easily extended to these tasks.
>
> Regardless of the possible extension of Metrix to other tasks, metric learning in itself in an important problem in machine learning with many applications. Since Metrix is an improvement in deep metric learning, our work does not have limited scope. In fact, since we bring in a new ingredient into improving deep metric learning, the scope of our work is more general than, for example, a new loss function.
>
> [1*] Hou et al., Learning a unified classifier incrementally via rebalancing, CVPR 2019.
>
> [2*] Snell et al., Prototypical networks for few-shot learning, arXiv 2017.
>
> [3*] Vinyals et al., Matching networks for one shot learning, NIPS 2016.
>
> **[Experiments]**
>
> Visualization of input mixup images are identical to that of Zhang et al., 2018. We will include tsne plots and some images of the retrieval.

---

> > ### Comment · Reviewer_S9kx · 2021-11-19
> > **Limited Scope**
> >
> > I thank authors for addressing some of my concerns. Its not clear to me how the proposed approach can be applied to other techniques like few-shot learning. [1*, 2*] are metric learning based few-shot techniques but it uses the absolute labels of the images for few shot learning whereas the proposed method (manly based on DML) uses the relative binary labels (+ve, -ve) for training the embedding. It is possible to generate the relative labels from absolute labels but you don't know how these types of labels would perform on few-shot learning. I don't think it would make sense to claim (in conclusions) that the method is generic and can be used for other types of problems like few-shot/ transfer learning problem.

---

> > > ### Author Response · Authors · 2021-11-22
> > > **Follow-up response to Reviewer S9kx**
> > >
> > > We thank the reviewer for acknowledging that we were able to *address some of their concerns*.
> > >
> > > We clarify that
> > >
> > > 1. [1*] uses relative labels, where for every anchor $x$, a positive is defined as the "embedding of the ground-truth class" and negatives are "embeddings of new classes that yield highest responses to $x$".
> > > 2. [2*] modifies absolute labels to relative labels as is standard in metric learning.
> > >
> > > We agree that it is not known how this extension would perform on few-shot learning. This is exactly why we mention it in the context of *future work*.
> > >
> > > We do not claim that "our method is generic and can be used for other types of problems like few-shot...". We rather claim that "our formulation is generic and it can be applied to a large class of loss functions". We use "generic" in the context of our formulation and not the tasks.
> > >
> > > We also claim that "our work *may* have application to few-shot learning..." and believe that this statement is plausible, since works like [1*,2*] use loss functions from metric learning. An extension to these tasks may not be trivial, but is plausible as future work. Although we believe that future work from the authors provides useful insights to the community, we are happy to remove this sentence if the reviewer insists.
> > >
> > > **[update on experiments]**
> > > Concerning the reviewer's original comment, we also updated Section B.3 in the Appendix with some qualitative results of the retrieval and the visualization of the embedding space on CUB200.

---

> > > > ### Comment · Reviewer_S9kx · 2021-12-01
> > > > **Final Rating**
> > > >
> > > > I thank authors for addressing most of my comments (3, 4,and 5). The modified version looks good. So I am changing my rating to 6. Regarding my concerns about few-shot learning, I think it is okay to mention it in the context of future wok.

---

### Official Review · Reviewer_LaZ6 · 2021-11-05

**Correctness:** 3
**Technical Novelty And Significance:** 3
**Empirical Novelty And Significance:** 3
**Recommendation:** 6
**Confidence:** 2

**Details Of Ethics Concerns:**

Not applicable.

**Main Review:**

The paper proposes a mixup strategy to augment the data, interpolate the labels and design a new loss function for deep metric learning. Overall, the paper is well written and easy to follow. And the results are improved compared with other models. 

I am not an expert in this area so the following concerns might not be important but I wish the authors could clarify. From my point of view, the results are impressive. The datasets used in the experiment are also popular enough for supporting the claims.

However, there are two concerns:

1) Although the authors propose the new general mixup loss function, the key elements of mixup from the paper follows from Zhang et al., 2018; Verma et al., 2019 and Venkataramananetal.,2021. In particular, the authors mentions "the idea of interpolating target labels is not straightforward." However, it seems that label interpolation is natraully derived from Zhang et al., 2018 in the paper. Could the authors please elaborate the key contributions from the paper? Is it the new generic loss function for mixup?

2) The authors propose a utilization method to support the claim of exporing new space using L2 distance in embedding space, which is not very convicing to me since L2 distance hardly captures information at the distribution level. From my point of view, the lower of the utilization score does not necessarily mean the model is exploring new and meaningful space. So could the author please elaborate on what is the key reason of using L2 distance in this case?


**Summary Of The Paper:**

The paper mentions the missing of studying both metric loss function and data augmentation techniques for the metric learning problem and proposes to use a mixup strategy for the improvement.  The authors claim the better results over the state-of-the-art using the mixup and use a new metric (utilization) to claim their method is exploring new space.

**Summary Of The Review:**

The authors propose a new and generic loss function in mixup scenario for deep metric learning. The experiments on popular datasets mostly support the claims in the paper with the needs of explaining some of ideas mentioned in the paper.

---

> ### Author Response · Authors · 2021-11-18
> **Response to Reviewer LaZ6**
>
> We appreciate the Reviewer LaZ6's valuable feedback. We address the concerns as follows:
>
> **[Label interpolation - natural derivation]**
>
> In the Section 2 - Related Works, paragraph "Interpolation for pairwise loss functions", we describe prior works that attempt mixup in deep metric learning.  Embedding expansion (Ko \& Gu, 2020), HDML (Zheng et al., 2019) and symmetrical synthesis (Gu \& Ko, 2020) propose to interpolate only embeddings within the same class but not target labels. MoCHI (Kalantidis et al., 2020) interpolate embeddings but thresholds $\lambda$ to 0.5 limiting the mixed labels to negatives. While these methods attempt to extend mixup from classification to deep metric learning or self-supervised learning, they do not propose a natural extension. This is challenging because unlike classification, the loss functions used in metric learning are not additive over examples.
>
> The key contributions in this work are that we *derive* a natural extension of mixup from classification to deep metric learning and that we propose a generic way of representing and interpolating labels (eq. 9). This allows straightforward extension of any kind of mixup to deep metric learning for a large class of loss functions. We empirically validate this generic loss function for mixup on both pair-based and proxy-based losses (table 2).
>
> **[$L_2$ distance in utilization]**
>
> Following alignment and uniformity (Wang and Isola, 2020), we use $L_2$ distance in utilization to measure the exploration in the embedding space across two distributions instead of one.
>
> What we are considering here is the sum of over all queries in the test set of the minimum $L_2$ distance between each query and all embeddings (clean and mixed) from the train set. While the $L_2$ distance between only two elements does not capture information at distribution level, the sum over all elements does.
>
> Through utilization, we claim that the model is exploring new space closer to the query, but do not claim that it explores *meaningful* space.

---

### Official Review · Reviewer_EXEY · 2021-11-06

**Correctness:** 4
**Technical Novelty And Significance:** 3
**Empirical Novelty And Significance:** 4
**Recommendation:** 6
**Confidence:** 4

**Main Review:**

**Strengths**
1. The paper is very clearly written and easy to follow. I really like the part on building the loss function in (9) step-by-step.
2. I really appreciate the extensive experiments that have been done in the paper.
3. The generalization of the loss function seems to be pretty standard, but the simple solution turns out to be very effective.

**Weaknesses**
1. My main concern is in terms of the error function in (10). Is there any explanation why the first clean loss must be there? What if only the mixed loss term is used as an objective function.
2. I appreciate the authors’ effort on the analysis in section 3.6 although I am not convinced what is the message we can get from (11). Moreover, the objective function also has the clean loss term and the analysis is only in terms of the mixed loss.

Minor Issue
1. In Section 4.1 in mixup settings, M(a) only uses positive-negative and anchor-negative pairs, while in section 3.5 in the description of M(a) positive-positive pairs are also defined as a subset of M(a), would it hurt the experiment results if positive-positive pairs are also used?
2. In the utilization section, the word "significantly" is used which is very dangerous in statistics. The uncertainty is not quantified, therefore there isn't enough evidence to use this word.


**Summary Of The Paper:**

The paper proposes a generalized loss function for the purpose of using mixup in deep metric learning, which extends the some existing loss functions for deep metric learning without mixup. Extensive experiments are conducted to show the superior performance of the proposed method.


**Summary Of The Review:**

The paper is overall very clearly written and easy to follow, the authors made a great effort to conduct the experiments in the paper. I am not an expert in this area and may have missed some related work in this area. My main concern is in terms of the error function in (10) as described in the main review.

---

> ### Author Response · Authors · 2021-11-18
> **Response to Reviewer EXEY**
>
> We appreciate the Reviewer EXEY's valuable feedback. We address the concerns as follows:
>
> **[Equation 10]**
>
> We appreciate the reviewer's question. In our initial experiments, we trained Metrix without the clean loss in eq. 10. We observe that on CUB200 using contrastive loss with embedding size $d=512$, R@1 drops from 64.7 (clean loss only) to 54.4. By using both clean and mixed loss, the performance improves from 64.7 to 67.4.
>
> We hypothesize that by removing the clean loss in eq. 10, we lose additional information that the network requires during learning. Here, we use mixup for data augmentation and not to replace the original data.
>
> We will add this experiment and discussion in the paper.
>
> **[Message from equation 11]**
>
> Clean embeddings are either positive or negative. While their contribution to the loss may depend on their distance from the anchor, their labels are still binary (+1 or -1). Their behaviour is known and needs no analysis.
>
> In Equation 11, we analyze the behaviour of mixed embeddings which has been under-studied. What we analyze is whether the contribution of these mixed embeddings towards the loss is positive, negative or in-between and whether this depends on is $\lambda$, because $\lambda$ determines the amount of mixing. We find that positivity is an increasing function of $\lambda$, as expected.
>
> We would be grateful if the reviewer could indicate what is missing or what is unclear with this approach.
>
> **[Using positive-positive pairs]**
>
> We performed our initial experiments by uniformly choosing at random between positive-positive, positive-negative and anchor-negative. We observed that with Metrix/feature on Cars-196 using Contrastive loss with embedding size $d=512$, R@1 dropped from 85.1 to 82.3. The influence of using positive-positive pairs only showing that it hurts the performance is also shown in the Ablation (Table 6), where R@1 drops from 85.1 to 81.1 on Cars-196.
>
> **[Use of word "significantly"]**
>
> We appreciate the reviewer's comment and thank them for pointing it out. We will remove the word ``significantly" in the utilization paragraph and replace it with synonyms elsewhere

---

### Official Review · Reviewer_gPed · 2021-11-07

**Correctness:** 3
**Technical Novelty And Significance:** 3
**Empirical Novelty And Significance:** 3
**Recommendation:** 8
**Confidence:** 4

**Main Review:**

Pros:
1) This is a good paper which has techincal novelty  - generalized loss for metrics learning, which allows to incorporate mixap augmentation. The authors also illustrate the effectiveness of the suggested technique for various losses previously used for Metric Learmning(Table 2).
2) The paper also suggest theoretical analysis of the "positivity", which is used for loss definition, (Section 3.6 and Figure 1) which aligns with empirical results.
3) The paper also performs comparison to different mixup methods and shows its superiority (Table 3)


Cons:
1) The paper is hard to read because many important formulations are found in the appendix (e.g evaluation used for Table 2 and Table 3 Recall @ K, and what K stands for is only explained in appendix B1, the discussion about influence of the approach on the training time is only found in B3,  a more eleborative discussion on positivity is presented in A2 and there are more examples). So the paper might benefit from  substantial revision or maybe  a resubmission to a journal might be considered.
2) The paper introduces new evaluation metric, utilization, validating that a representation more appropriate for test classes is implicitly learned during exploration of the embedding space in the presence of mixup. What is not clear to me is if this metrics is measured on the classes that particiapte in training (After eq.12 -  Utilization measures the average, over the test set Q, of the minimum distance of a query q to a training example x ∈ X in the embedding space of the trained model f (lower is better)). If this is the case this metrics is not really interesting with respect to the actual goal of metrics learning (to be able to generalize for unseen classes) -- it would be helpful if the authors can clarify this point and also elaborate why this metrics is more interesting than other available metrics.
3) Minor issue: In Table 4 SOP dataset - how can we get 5 examples per class if there are 22, 634 classes and 60, 026 images in training (so if we assume balanced dstribution of classes we will get ~2.65 images per class) -- are you using the same images?
4) Are you normalizing embeddings ?




**Summary Of The Paper:**

The paper deals with the problem of metrics learning. It proposes to extend Mixap data augmentation from calssification to metrics learning. This Mixap data augmentation approach interpolates two or more examples (either directly in the input or the corresponding embeddings - eq.4) and corresponding target labels at a time. This task is challenging because unlike classification, the loss functions used in metric learning are not additive over examples and hence the authors claim that the idea of interpolating target labels is not straightforward.
The authors suggest a general formulation to accomodate mixup for many losses currenly used in metrics learning (see eqs. 3, 7, 9, 10 for generic formulation  and  eqs. 13, 14,15 for formulation of specific losses). The authors provie extensive experiments that show the effectiveness of their techqniue (Table 2)  on 4 different dataset (Table 4)



**Summary Of The Review:**

I recommend to accept this paper because I think that the paper is relevant to the ICLR community and has some innovation supported by experiments (and some minor theory).  It also serves as a good survey for the community with many useful details for AI practicionaries in the field of metric learning, which are outlined in appendix. However, my concern is that the paper is not well organized and maybe better suited for a journal venue.

---

> ### Author Response · Authors · 2021-11-18
> **Response to Reviewer gPed**
>
> We appreciate the Reviewer gPed's valuable feedback. We address the concerns as follows:
>
> **[Paper organization]**
>
> Considering the page limit and the amount of material in our paper, we prioritized the items. In this sense, we  follow one of the reviewer's suggestion but not all:
> 1. We will include the details of training time in the main paper.
> 2. Recall@K is not our contribution and is a commonly used evaluation metric in deep metric learning defined by Oh Song et al., 2016.
> 3. The proof of positivity (Section A.2) is not necessary for following the paper.
>
> Reviewers EXEY, LaZ6 and c8hM acknowledge that the paper is *very clearly written and easy to follow*.
>
> **[Utilization]**
>
> We compute utilization between the query $q$ (test set) and the training set $X$ where the classes in test and training set are *different*. It is a standard practice in metric learning that classes at inference are unseen during training. We clarify this throughout the paper (4th paragraph of Section 1, Section 4.3 and Section 5).
>
> The exploration of the embedding space by the network using utilization measures the extent to which the embedding of the test example is closer to any embedding in the training set. This shows that a representation more appropriate for test classes is implicitly learned during exploration of the embedding space in the presence of mixup.
>
> Standard metrics like Recall@K is used as an evaluation metric only during inference i.e. on the test distribution. Alignment and Uniformity (Wang and Isola, 2020) are limited to a single distribution or dataset, either the training set (as loss functions) or the test set (as evaluation metrics). Utilization measures the similarity across two distributions (train and test) rather than one.
> Thus, utilization gives us additional quantitative measurement about *why* our method works better.
>
> **[5 examples per class in SOP]**
>
> SOP dataset has 22,634 classes with 120,053 images. The first 11,318 classes (60,026 images) are used for training while the other 11,316 (60,027 images) classes are used for testing. Hence there are approximately 5.3 images per class in the training set. We will clarify in the Appendix.
>
> **[Normalizing embeddings]**
>
> Yes, we normalize embeddings

---

### Author Response · Authors · 2021-11-23
**Summary of revisions**

We thank all the reviewers for their time and valuable feedback. Following their suggestions, we have made the following changes to the manuscript:

1. We provide an illustration of the exploration of the embedding space, with and without mixed examples in Section A.3 of the Appendix (Figure 3).
2. We provide some qualitative results of retrieval in Section B.3 of the Appendix (Figure 4).
3. We provide the embedding space visualization of CUB200 test examples in Section B.4 of the Appendix (Figure 5).
4. We added the definition of Recall@K in Section 4.1 - paragraph "Methods" (page 7).
5. We modified the last sentence of Section 3.6.

---

### Decision · Program_Chairs · 2022-01-20

**Decision:**

Accept (Poster)

**Comment:**

This paper adapts the mixup data augmentation strategy to the case of metric learning. The main challenge addressed is the fact that in metric learning, the loss function does not treat each example as an IID sample. The paper takes the view of metric learning as learning over positive and negative pairs (those belonging to the same/different classes) and uses this to develop a fairly general metric-mixup formulation. To measure the effectiveness of the approach for metric learning, the paper introduces a new measure called utilization that looks at the distance of a query point to its nearest training point in embedding space.

The reviewers (5 of them) all favour acceptance on the grounds of novelty, and the performance of the method. During the discussion, some issues were raised around whether utilization is a useful measure, improvements to the paper clarity, whether the clean loss in eq. 10 is necessary, and potential limitations on the generality of the approach. However, additional experiments and clarification during the discussion period has resolved these issues to their satisfaction.